# Personalized Prediction By Learning Halfspace Reference Classes Under Well-Behaved Distribution

## Abstract

In machine learning applications, predictive models are trained to serve future queries across the entire data distribution. Real-world data often demands excessively complex models to achieve competitive performance, however, sacrificing interpretability. Hence, the growing deployment of machine learning models in high-stakes applications, such as healthcare, motivates the search for methods for accurate and explainable predictions. This work proposes a *personalized prediction* scheme, where an easy-to-interpret predictor is learned per query. In particular, we wish to produce a *sparse linear* classifier with competitive performance specifically on some sub-population that includes the query point. The goal of this work is to study the PAC-learnability of this prediction model for sub-populations represented by *halfspaces* in a label-agnostic setting. We first give a distribution-specific PAC-learning algorithm for learning reference classes for personalized prediction. By leveraging both the reference-class learning algorithm and a list learner of sparse linear representations, we prove the first upper bound, $O(\mathrm{opt}^{1/4})$, for personalized prediction with sparse linear classifiers and homogeneous halfspace subsets. We also evaluate our algorithms on a variety of standard benchmark data sets.

## 1 Introduction

In real-world machine learning applications, complex models, such as deep neural networks and transformers, are often preferred than simpler models, such as linear classifiers, due to their ability to achieve higher predictive accuracy. However, relying on models that perform well on average across the entire populations introduces a dilemma: expressivity is often at odds with interpretability. For instance, a doctor assessing the safety of a medication for a patient needs to understand the factors influencing a model's "safe" prediction before proceeding with treatment. Similarly, an investor allocating substantial funds would require insight into the reasoning behind a model's investment recommendations. Overall, the opaqueness of the prediction process of complex machine learning models can often hinder trust and adoption in high-stakes applications (Qi et al., 2018; Rudin, 2019).

Despite that interpreting the behaviors of complex models has been widely studied (Ribeiro et al., 2016; Lundberg and Lee, 2017; Ribeiro et al., 2018; Wang and Wang, 2021), these methods either interpret the local behaviors by simple models or (approximately) estimate certain statistics that assist interpretation of relevant properties. Huang and Marques-Silva (2024) demonstrated that these "post hoc" methods for explaining the prediction behaviors of complex models could be misleading in high-stake applications, which motivates the usage of *inherently* interpretable models, i.e., models themselves are explanations. Unfortunately, in the real world, easy-to-interpret rules, such as conjunctions and linear representations, are often too simple to accurately capture the properties we care about across the entire population.

In this work, a personalized prediction scheme is adopted to reconcile model interpretability with performance by learning distinct models for different observations. Specifically, for every query point, we seek a simple decision rule along with a sub-population which not only includes the query point, but is captured accurately by the simple rule. The appeal of such an approach is clear in applications where interpretability (of the classifier) is needed. Such settings include, e.g., medical diagnosis and bioinformatics (Khan et al., 2001; Hanczar and Dougherty, 2008). In particular, we

study the *distribution-specific* PAC-learnability of sparse linear classifiers on subsets defined by homogeneous halfspaces in the personalized prediction scheme, in the presence of *adversarial* label noise or *agnostic* setting (Kearns et al., 1994).

## 1.1 BACKGROUND

The need for *personalization* has emerged in a variety of machine learning application areas, e.g., cognitive science (Fan and Poole, 2006), recommendation systems (Zhang et al., 2020; McAuley, 2022), disease diagnosis (Finkelstein and Jeong, 2017) and treatment (Lipkovich et al., 2017), medical device development (Lee et al., 2020), patient care (Golany and Radinsky, 2019), etc. Various techniques have been developed to endow machine learning models with personalized behaviors. Early methods for personalization (Linden et al., 2003) made significant achievements in a variety of commercial applications, such as search engines (Pretschner and Gauch, 1999; Speretta and Gauch, 2005) and recommendation systems (Resnick and Varian, 1997; Shani and Gunawardana, 2011). These approaches inherently limited the choice of representations usable as predictors, and fell short in interpretability. In applications that could impact human health and welfare, personalization is often achieved by incorporating techniques such as feature engineering (Finkelstein and Jeong, 2017; Schneider and Handali, 2019; Lee et al., 2020), group-attribute-based or heuristic-based data clustering (Taylor et al., 2017; Lipkovich et al., 2017; Bertsimas et al., 2019; Schneider and Handali, 2019; Schneider and Vlachos, 2020), or data re-weighting (Schneider and Vlachos, 2020) into the existing training processes of various machine learning models. These methods aim to increase the number of training examples for each individual either by assuming multiple examples per person or finding a "similar" subgroup based on some predetermined heuristic distance measure, which potentially requires expert knowledge. More recently, due to the tremendous success of Large Language Models, much effort has been invested into model alignment for personalization (Jang et al., 2023; Chen et al., 2025), but without focus on interpretability.

Although much progress has been made in personalizing prediction, little attention been paid to making these predictions interpretable, and there has been no theoretical analysis of the performance. In this work, we propose a *personalized prediction* (cf. Definition 2.1) scheme to address these problems, specifically for *binary classification* tasks.

**Personalized Prediction:** Instead of learning a universal classifier to predict all future queries, we learn a dedicated classifier for each incoming query to predict exclusively on it. The key difference between our learning scheme and the standard one is that we only model a subset of the whole data population, which well represents the incoming query. That is, we jointly learn a classifier and a subset such that not only the members in the subset resembles the query point in some reasonable measure, but also the classifier performs better on the subset than on the whole population. In this work, we only consider the class of subsets characterized by *homogeneous* halfspaces[1] for computational reasons that will be elaborated in Section 2.2.

**Interpretability:** We consider the class of classifiers to be $s$-sparse linear classifiers, which are linear classifiers with at most $s$ *non-zero* weights, for $s = O(1)$. In practice, we typically take $s \approx 2$ so that a human can understand the decision process.

Again, the *intuition* behind personalized prediction is that the underlying property of a sub-population is likely easier to capture by simple representation classes than that of the entire distribution. This belief is supported by real-world evidence from several sources: Rosenfeld et al. (2015) showed that within a certain sub-population, the risk of gastrointestinal cancer is strongly correlated with some attributes that are not predictive in general. Izzo et al. (2023), Hainline et al. (2019), and Calderon et al. (2020) demonstrated that linear regression on a portion of the data may perform as well as more complex models learned on the full dataset in many standard real-world benchmarks.

## 1.2 OUR RESULTS

**PAC-learnability:** Our main contribution is the first PAC-learning algorithm for personalized prediction (cf. Definition 2.1) with *sparse* linear classifiers as predictors and homogeneous halfspace as subsets. We proved a $O(\text{opt}^{1/4})$ upper bound (cf. Theorem 4.2) for our main algorithm (cf. Algorithm 3) under distributions with *well-behaved* attribute marginals (see Appendix C for details).

---

[1]A halfspace can be defined as the set of all points on one side of a hyperplane. See Section 2.1 for details.

**Experiments:** We empirically evaluated our algorithm on multiple standard UCI medical datasets. For these benchmarks, both the need for interpretability and the relatively small data size strongly motivate the use of sparse classifiers. We compared the accuracy of the personalized predictions to the accuracy of a sparse ERM for each data set, and found that it is generally much higher, on par with less-interpretable standard classification methods such as logistic regression and SVM.

**Organization:** In Section 2, we introduce the necessary mathematical notations, and discuss the computational challenges of personalized prediction with subsets as halfspaces. In Section 3, we present our algorithms for learning reference classes. In Section 4, we present our personalized prediction algorithm, which uses the reference class learning algorithm as a subroutine, and show our empirical evaluationon several UCI datasets. At last, we discuss our limitation and future directions.

### 1.3 TECHNICAL OVERVIEW

Overall, the core of our approach is a *projected* gradient descent (PGD) algorithm (cf. Algorithm 2) for *learning reference classes* (cf. Definition 2.2). Briefly, learning reference class is essentially equivalent to the personalized prediction problem if the class of classifiers given in personalized prediction only consists of a single classifier (see Section 2.2). If we can learn reference class, we are able to solve the personalized prediction problem with any finite class of classifiers by enumerating the class of classifiers. Following Huang and Juba (2025), we observe that an algorithm (cf. Algorithm 4) for *robust list learning* (cf. Definition 4.1) may be leveraged to perform personalized prediction for large or infinite classifier classes, such as sparse linear classifiers, by reducing them to finite sets.

Our performance analysis of PGD is inspired by Huang and Juba (2025), who was solving the *conditional classification* problem. The problem is similar to personalized prediction in the sense that it also seeks for a classifier with small classification loss on some jointly learned subset, but differs in a key way that the subset is not required to contain any point. They employed a different projected gradient descent method, whose convergence implicitly implies optimality due to the observation that the projected gradient always approximately points to the optimal solution. However, their reasoning does not necessarily hold if we modify their algorithm to ensure we end up with a subset containing the query point. Like them, we are able to utilize the same property, but we use it rather differently: inspired by Diakonikolas et al. (2022), we find that PGD decreases the distance between its hypothesis and the optimal solution by this property, and this closeness in distance can be translated to closeness in loss. Within this distance-based analysis, the membership of the query point can be secured without increasing the distance (or loss) by a contractive projection. We stress that we proved the property (cf. Lemma 3.2) mentioned above under the more general well-behaved family as oppose to Gaussian distributions assumed in Huang and Juba (2025), however, with slightly worse guarantee.

### 1.4 RELATED WORKS

A related line of work, conditional learning (Juba, 2017; Calderon et al., 2020; Hainline et al., 2019; Liang and Juba, 2022; Huang and Juba, 2025), typically incorporates two sub-problems, obtaining a finite list of predictors, learning a predictor-subset pair out this finite list and some class of subsets. Many algorithms for "list-decodable" learning (Definition 4.1) to obtain a list of predictors have been proposed (Charikar et al., 2017; Kothari et al., 2018; Calderon et al., 2020; Bakshi and Kothari, 2021; Liang and Juba, 2022). The latter problem was reduced to the problem of learning abduction Juba (2016a): formally, this is the problem of learning a subset of the data distribution where e.g., no errors occur. In their work, they showed that subsets defined by $k$-DNFs can be efficiently learned in realizable cases without any distributional assumptions. Subsequent improvements were obtained for the agnostic setting (Zhang et al., 2017; Juba et al., 2018). Juba (2016a; 2017) and Durgin and Juba (2019) observed one-sided learning of conjunctions leads to a computational barrier in the distribution-free setting, hence the focus on $k$-DNF subsets in those works.

Learning mixtures of sparse models is a topic seemingly related to our problem. Various problems were studies under this topic, some were trying to learn multiple sparse linear models when given model responses (Gandikota et al., 2020; Polyanskii, 2021), others were focusing on mean recovery with sample access to unknown mixture of sparsely parameterized distributions (Pal and Mazumdar, 2022; Mazumdar and Pal, 2024). However, these works were usually conducted in noise-free settings. Recall that the representation class we are considering is a combination of sparse linear predictors and halfspaces, whose classification error is only measured on one side of the halfspaces. If, off the

support of the reference class, the distribution is not modeled well by a mixture of classifiers, then there is no guarantee on the quality of the "personalized" prediction we would obtain. Thus, our objective is not captured by learning mixtures of sparse classifiers.

# 2 PRELIMINARIES

## 2.1 MATHEMATICAL NOTATIONS

In general, we use lowercase italic font characters to represent scalars, e.g. $x \in \mathbb{R}$, lowercase bold italic font characters to represent vectors, e.g. $\boldsymbol{x} \in \mathbb{R}^d$. In particular, subscripts will be used to index the coordinates of any vector, e.g., $x_i$ represents the $i$th coordinate of the vector $\boldsymbol{x}$. For random variables, we use lowercase normal font characters to represent random scalars, e.g. $\mathrm{x} \in \mathbb{R}$, and lowercase bold normal font characters to represent random vectors, e.g. $\mathbf{x} \in \mathbb{R}^d$. For $\mathbf{x} \in \mathbb{R}^d$, let $\|\mathbf{x}\|_p = (\sum_{i=1}^d |x_i|^p)^{1/p}$ denote the $l_p$-norm of $\mathbf{x}$, and $\bar{\mathbf{x}} = \mathbf{x}/\|\mathbf{x}\|_2$ denote the normalized vector of $\mathbf{x}$. We will use $\langle \mathbf{x}, \boldsymbol{u} \rangle$ to represent the inner product of $\mathbf{x}, \boldsymbol{u} \in \mathbb{R}^d$, $\mathbf{x}^{\otimes k}$ to represent the outer product of $\mathbf{x} \in \mathbb{R}^d$ to the $k$th degree, and $\theta(\boldsymbol{u}, \boldsymbol{w})$ to denote the angle between two vectors $\boldsymbol{u}, \boldsymbol{w} \in \mathbb{R}^d$.

For any subspace $V \subseteq \mathbb{R}^d$, let $\mathbf{x}_V$ denote the projection of $\mathbf{x}$ onto $V$. Further, we will write $\boldsymbol{w}^\perp = \{\boldsymbol{u} \in \mathbb{R}^d \mid \langle \boldsymbol{u}, \boldsymbol{w} \rangle = 0\}$ as the orthogonal space of $\boldsymbol{w} \in \mathbb{R}^d$, and, therefore, $\mathbf{x}_{\boldsymbol{w}^\perp} = (I - \bar{\boldsymbol{w}}^{\otimes 2})\mathbf{x}$ as the projection of $\mathbf{x} \in \mathbb{R}^d$ onto $\boldsymbol{w}^\perp$. For subsets of $\mathbb{R}^d$, let $S_1 \cap S_2$ be the intersection of $S_1, S_2$ and $S_1 \cup S_2$ be the union of $S_1, S_2$. Meanwhile, we denote $S_1 \backslash S_2 = \{\mathbf{x} \in \mathbb{R}^d \mid \mathbf{x} \in S_1, \mathbf{x} \notin S_2\}$ and $S^c = \{\mathbf{x} \in \mathbb{R}^d \mid \mathbf{x} \notin S\}$ for the set complement operation.

For probabilistic notation, we use $\mathcal{D}_{\mathbf{x}}$ to denote the 1-dimensional marginal distribution of $\mathcal{D}$ on the direction $\mathbf{x} \in \mathbb{R}^d$, $\mathrm{Pr}_{\mathbf{x} \sim \mathcal{D}}\{\mathbf{x} \in S\}$ to denote the probability of an event $\mathbf{x} \in S$, $\mathbb{E}_{\mathbf{x} \sim \mathcal{D}}[f(\mathbf{x})]$ to denote the expectation of some statistic $f(\mathbf{x})$, and therefore, $\hat{\|}f(\mathbf{x})\hat{\|}_p = (\mathbb{E}_{\mathbf{x} \sim \mathcal{D}}[\|f(\mathbf{x})\|_p^p])^{1/p}$. In particular, for an i.i.d. sample $\hat{\mathcal{D}} \sim \mathcal{D}$, we define the empirical probability and expectation as $\mathrm{Pr}_{\mathbf{x} \sim \hat{\mathcal{D}}}\{\mathbf{x} \in S\} = \frac{1}{|\hat{\mathcal{D}}|}\sum_{\mathbf{x} \in \hat{\mathcal{D}}} \mathbb{1}\{\mathbf{x} \in S\}, \quad \mathbb{E}_{\mathbf{x} \sim \hat{\mathcal{D}}}[f(\mathbf{x})] = \frac{1}{|\hat{\mathcal{D}}|}\sum_{\mathbf{x} \in \hat{\mathcal{D}}} f(\mathbf{x})$. For simplicity of notation, we may drop $\mathcal{D}$ from the subscript when it is clear from the context, i.e., we may simply write $\mathrm{Pr}\{\mathbf{x} \in S\}, \mathbb{E}[f]$ for $\mathrm{Pr}_{\mathbf{x} \sim \mathcal{D}}\{\mathbf{x} \in S\}, \mathbb{E}_{\mathbf{x} \sim \mathcal{D}}[f]$.

We define **halfspaces** as subsets of $\mathbb{R}^d$ as follows. For any $t \in \mathbb{R}$ and $\boldsymbol{w} \in \mathbb{R}^d$, a $d$-dimensional halfspace with threshold $t$ and normal vector $\boldsymbol{w}$ is defined as $h_t(\boldsymbol{w}) = \{\mathbf{x} \in \mathbb{R}^d \mid \langle \mathbf{x}, \boldsymbol{w} \rangle \geq t\}$ (resp. $h_t^c(\boldsymbol{w}) = \{\mathbf{x} \in \mathbb{R}^d \mid \langle \mathbf{x}, \boldsymbol{w} \rangle \leq t\}$). For homogeneous halfspaces ($t = 0$), we write $h(\boldsymbol{w})$ for $h_0(\boldsymbol{w})$.

## 2.2 PERSONALIZED PREDICTION AND COMPUTATIONAL CHALLENGES

Motivated by the observation (at the end of Section 1.1) that different populations may have different population-specific risk factors, we consider the following definition of a personalized prediction problem. In this problem, our algorithm is given the attributes of a specific individual that we would like to make a prediction about. The algorithm searches for the population that individual belongs to that yields the most accurate sparse classifier, to use to make our prediction for the individual.

**Definition 2.1** (Personalized Prediction). *Let $\mathcal{D}$ be any probability distribution over $\mathbb{R}^d \times \{0, 1\}$, $\mathcal{C} \subseteq \{c : \mathbb{R}^d \rightarrow \{0, 1\}\}$ be a class of classifiers, and $\mathcal{H}$ be a collection of subsets of $\mathbb{R}^d$. For parameters $\alpha > 0$ and $\epsilon, \delta \in (0, 1)$, the $\alpha$-approximate Personalized Prediction problem is, given $m$ labeled examples drawn from $\mathcal{D}$ and a query point $\boldsymbol{x}' \in \mathbb{R}^d$, to return a pair $(c, S) \in \mathcal{C} \times \mathcal{H}$ with $\boldsymbol{x}' \in S$ such that with probability $1 - \delta$, for any $(c^*, S^*) \in \mathcal{C} \times \mathcal{H}$ with $\boldsymbol{x}' \in S^*$,*

$$\mathrm{Pr}_{(\mathbf{x}, \mathrm{y}) \sim \mathcal{D}}\{c(\mathbf{x}) \neq \mathrm{y} \mid \mathbf{x} \in S\} \leq \alpha \mathrm{Pr}_{(\mathbf{x}, \mathrm{y}) \sim \mathcal{D}}\{c^*(\mathbf{x}) \neq \mathrm{y} \mid \mathbf{x} \in S^*\} + \epsilon.$$

*If $\alpha = 1$, we simply refer to the problem as Personalized Prediction.*

As discussed, we choose $\mathcal{C}$ to be sparse linear classifiers for interpretability. Thus, the choice of $\mathcal{H}$ is crucial for PAC-learnability. Typically, $\mathcal{H}$ is supposed to satisfy some population lower bound, i.e., $\mathrm{Pr}\{\mathbf{x} \in S\} \geq \mu$ for every $S \in \mathcal{H}$ and some constant $\mu \in (0, 1)$, because otherwise one can easily construct trivial solutions, such as a singleton $S^*$, to make the selected subsets statistically meaningless. As the first attempt to obtain a distribution-specific PAC-learning guarantee for agnostic personalized prediction, we choose to work with halfspace (subsets), since its distribution-specific

Table 1: upper and lower bounds for halfspaces in $\text{poly}(d, 1/\text{opt})$ time for different tasks.

| Task | Halfspace Type | Distribution | Upper Bound | Lower Bound |
|---|---|---|---|---|
| Classification | General | Gaussian | $O(\text{opt})$ | $\text{opt} + \Omega(1/\sqrt{\log d})$ |
| Classification | Homogeneous | Well-behaved | $O(\text{opt})$ | N/A |
| Conditional Classification | General | Gaussian | N/A | $\text{opt} + \Omega(1/\sqrt{\log d})$ |
| Conditional Classification | Homogeneous | Gaussian | $\tilde{O}(\sqrt{\text{opt}})$ | N/A |

PAC-learnability is well studied (Diakonikolas et al., 2020b;c; 2021; 2022; 2024). Even so, it is still difficult to learn (under Definition 2.1) this relatively simple class without further restrictions.

Without distributional assumptions, it is computationally challenging to achieve even a much weaker version of personalized prediction with $\mathcal{H}$ to be halfspaces. Suppose, in Definition 2.1, the classifier class consists of a single classifier that makes no error on some subset in the subset class, then personalized prediction is equivalent to learning a *reference class* (Juba, 2016b; Hainline et al., 2019).

**Definition 2.2** (Reference Class). *Let $\mathcal{D}$ be any probability distribution over $\mathbb{R}^d \times \{0, 1\}$ and $\mathcal{H}$ be a collection of subsets of $\mathbb{R}^d$. For parameters $\epsilon, \delta \in (0, 1)$, the Reference Class learning problem is, given $m$ labeled examples drawn from $\mathcal{D}$ and a query point $\boldsymbol{x}' \in \mathbb{R}^d$, to return a subset $S \in \mathcal{H}$ with $\boldsymbol{x}' \in S$ such that $\Pr_{(\mathbf{x}, \mathrm{y}) \sim \mathcal{D}}\{\mathrm{y} = 1 \mid \mathbf{x} \in S\} \geq 1 - \epsilon$ with probability $1 - \delta$.*

Unfortunately, Juba and Li (2020) showed that any $\mathcal{H}$ with the ability to express *conjunctions* (ANDs of Boolean literals) cannot be efficiently learned as a reference class. As halfspaces may express conjunctions on $\{0, 1\}^d$ domain, personalized prediction with halfspace subsets is intractable without distributional assumptions, even in the noiseless setting. Therefore, in the presence of adversarial noise, the use of some niceness assumptions on the attribute marginals seems inevitable.

Despite the simplicity of halfspaces in comparison to models, such as neural networks and transformers, it is surprisingly challenging to obtain a descent upper bound for agnostically learning halfspaces even under nice distributions. On the other hand, a recent work by Diakonikolas et al. (2023) presented a distribution-specific *cryptographic* lower bound for learning halfspaces as shown in Table 1. Of greater relevance, Huang and Juba (2025) proved a similar lower bound (see Table 1) for *conditional classification* (cf. Definition A.1), which resembles personalized prediction in many ways. In fact, we prove that personalized prediction is at least as hard as conditional classification.

**Claim 2.3** (Informal). *Conditional classification is efficiently reducible to personalized prediction.*

Therefore, the lower bound for conditional classification shown in Table 1 suggest potential computational barriers for learning general halfspace subsets in personalized prediction even under Gaussian distributions. Other problems with a similar structure, which require models of sub-populations defined by halfspaces, often exhibit comparable or even stronger hardness (Hsu et al., 2024). These observations motivate us to consider personalized prediction with a subset class that is strictly simpler than general halfspaces, i.e., homogeneous halfspaces, under nice distributions.

## 3 LEARNING OF HOMOGENEOUS HALFSPACE REFERENCE CLASS

In this section, we present our learning algorithms for homogeneous halfspaces reference classes under distributions with well-behaved $\mathbf{x}$-marginals (see Appendix C for formal definitions). Noticeably, these algorithms will be used as subroutines in the Algorithm 3 introduced in Section 4.

**Well-Behaveness:** Informally speaking, the family of *well-behaved distributions* must satisfy the following properties: every low-dimensional marginal of a the distribution must have sub-exponential tail, density bounds, low-degree moment upper bounds, and every halfspace containing the distribution mean must have non-negligible probability mass. The well-behaved family is a natural generalization of many common distributions, such as uniform, Gaussian, and many log-concave distributions (Lovász and Vempala, 2007; Diakonikolas et al., 2020c). For completeness, we prove a few instances in Appendix C. Note that the parameters of these distributional properties only matters in proving the fully parameterized theorems presented in the appendix. For better clarity, we suppress the distribution related parameters in the main paper, as they won't affect our guarantees asymptotically.

While directly optimizing the target loss $\Pr\{y = 1 \mid \mathbf{x} \in h(\boldsymbol{w})\}$ is hard in general, Huang and Juba (2025) showed there exists a simple convex surrogate approximation to this kind of target loss that may approximately captures the optimal solution, i.e., $\mathcal{L}_{\mathcal{D}}(\boldsymbol{w}) = \mathbb{E}[y \cdot \max(0, \langle \mathbf{x}, \boldsymbol{w} \rangle)]$. Even though our objective functions are the same, we further require the resulting halfspace $h(\boldsymbol{w})$ to contain the query point $\boldsymbol{x}$. Interestingly, we show that a few tweaks on the gradient descent algorithms given in Huang and Juba (2025) can guarantee $\boldsymbol{x} \in h(\boldsymbol{w})$ with the same performance.

## 3.1 Algorithm Overview

Overall, Algorithm 1 consists of both pre-processing and post-processing for Algorithm 2, while Algorithm 2 is our main learning algorithm for homogeneous halfspace reference classes.

| **Algorithm 1** Learning Reference Class | **Algorithm 2** PGD With Contractive Projection |
|---|---|
| 1: **procedure** REFCLASS($\mathcal{D}, \epsilon, \delta, \boldsymbol{x}$) 
 2:     $T \leftarrow O(\epsilon^{-5/4})$ 
 3:     $\lambda \leftarrow O(\epsilon^{3/4})$ 
 4:     $\hat{\mathcal{D}}_1 \leftarrow \tilde{O}(\epsilon^{-1})$-sample from $\mathcal{D}$ with negated labels 
 5:     $\mathcal{W} \leftarrow$ PROJECTEDGD($\hat{\mathcal{D}}_1, T, \lambda, \boldsymbol{x}$) 
 6:     $\hat{\mathcal{D}}_2 \leftarrow \tilde{O}(\epsilon^{-1/2})$-sample from $\mathcal{D}$ 
 7:     $\boldsymbol{w}^* \leftarrow \max\limits_{\boldsymbol{w} \in \mathcal{W}} \Pr_{\hat{\mathcal{D}}_2}\{y = 1 \mid \mathbf{x} \in h(\boldsymbol{w})\}$ 
 8:     **return** $\boldsymbol{w}^*$ 
 9: **end procedure** | 1: **procedure** PROJECTEDGD($\hat{\mathcal{D}}, T, \lambda, \boldsymbol{x}$) 
 2:     $\boldsymbol{w}^{(0)} \leftarrow \bar{\boldsymbol{x}}$ 
 3:     **for** $i = 1, \ldots, T$ **do** 
 4:         $\boldsymbol{u}^{(i)} \leftarrow \boldsymbol{w}^{(i-1)} - \lambda \mathbb{E}_{\hat{\mathcal{D}}}[g_{\boldsymbol{w}^{(i-1)}}(\mathbf{x}, y)]$ 
 5:         **if** $\langle \boldsymbol{u}^{(i)}, \boldsymbol{x} \rangle < 0$ **then** 
 6:             $\boldsymbol{w}^{(i)} \leftarrow \bar{\boldsymbol{u}}_{\boldsymbol{x}_\perp}^{(i)}$ 
 7:         **else** 
 8:             $\boldsymbol{w}^{(i)} \leftarrow \bar{\boldsymbol{u}}^{(i)}$ 
 9:         **end if** 
 10:    **end for** 
 11:    **return** $(\boldsymbol{w}^{(0)}, \ldots, \boldsymbol{w}^{(T)})$ 
 12: **end procedure** |

Notably, the training set $\hat{\mathcal{D}}_1$ is sampled from $\mathcal{D}$ with *negated labels* because Algorithm 2 is designed to solve minimization problems. Negating the labels allows us to equivalently minimize $\Pr\{y = 0 \mid \mathbf{x} \in h(\boldsymbol{w})\}$ instead of maximizing $\Pr\{y = 1 \mid \mathbf{x} \in h(\boldsymbol{w})\}$. Given that Algorithm 2 returns a list of halfspaces, one of which is guaranteed to have $\Pr\{y = 1 \mid \mathbf{x} \in h(\boldsymbol{w})\} = 1 - O(\epsilon^{1/4})$, we sample a validation set $\hat{\mathcal{D}}_2$ to select a good halfspace from the list. Inspired by Huang and Juba (2025), our Algorithm 2 uses the projected gradient $g_{\boldsymbol{w}}(\mathbf{x}, y) = y \cdot \mathbf{x}_{\boldsymbol{w}^\perp} \mathbb{1}\{\mathbf{x} \in h(\boldsymbol{w})\}$ to update the normal vector $\boldsymbol{w}$. Also motivated by Diakonikolas et al. (2022), we show that our Algorithm 2 is guaranteed to return at least one good halfspace through an *angle contraction* analysis next.

## 3.2 Performance Analysis

We now state our main theorem for Algorithm 1, but postpone the formal proof to Appendix D. Notice that REFCLASS (cf. Algorithm 1) is actually no more than a wrapper of PROJECTEDGD (cf. Algorithm 2) with some empirical estimates. Therefore, we focus on analyzing Algorithm 2 here.

**Theorem 3.1.** *Let $\mathcal{D}$ be any distribution on $\mathbb{R}^d \times \{0, 1\}$ with centered well-behaved $\mathbf{x}$-marginal and $\boldsymbol{x} \in \mathbb{R}^d$ be an query. If there exists a unit vector $\boldsymbol{v} \in \mathbb{R}^d$ such that $\boldsymbol{x} \in h(\boldsymbol{v})$ and $\Pr\{y = 1 \mid \mathbf{x} \in h(\boldsymbol{v})\} \geq 1 - \epsilon$, then, with at most $\tilde{O}(\epsilon^{-1})$ examples, Algorithm 1 runs in time at most $\tilde{O}(d\epsilon^{-9/4})$ and returns a $\boldsymbol{w}^*$ such that $\boldsymbol{x} \in h(\boldsymbol{w}^*)$ and $\Pr\{y = 1 \mid \mathbf{x} \in h(\boldsymbol{w}^*)\} = 1 - O(\epsilon^{1/4})$ w.h.p.*

Prior to the detailed analysis, we sketch the main proof idea as follows. It can be shown that the *gradient step* (Line 4 of Algorithm 2) decreases the angle between the optimal normal vector $\boldsymbol{v}$ and the algorithm's "guess" $\boldsymbol{w}$ by a fixed amount in each iteration of Algorithm 2 as long as the halfspace $h(\boldsymbol{w})$ is far from optimal. This implies that, with a few iterations, the output of Algorithm 2 will contain at least one halfspace of low error. Then, we can use this guarantee of Algorithm 2 to show the optimality of Algorithm 1 with a simple label mapping and empirical risk estimation.

As a key property to ensure *angle contraction* for each gradient step, we observed that the projected gradient $\mathbb{E}[-g_{\boldsymbol{w}}(\mathbf{x}, y)]$ always approximately "points" at the right direction or, in another word, the projected gradient has non-negligible correlation with the optimal normal vector $\boldsymbol{v}$ if $\boldsymbol{w}$ is significantly sub-optimal. In particular, Huang and Juba (2025) proved the same property under Gaussian $\mathbf{x}$-marginals, we show that slightly worse guarantee holds under well-behaved $\mathbf{x}$-marginals.

**Lemma 3.2** (Gradient Projection Lower Bound). *Let $\mathcal{D}$ be any distribution on $\mathbb{R}^d \times \{0,1\}$ with centered well-behaved $\mathbf{x}$-marginal, and $g_{\boldsymbol{w}}(\mathbf{x}, y) = y \cdot \mathbf{x}_{\boldsymbol{w}^\perp} \mathbb{1}\{\mathbf{x} \in h(\boldsymbol{w})\}$. Suppose there exists a unit vector $\boldsymbol{v} \in \mathbb{R}^d$ that satisfies $\Pr\{y = 1 \mid \mathbf{x} \in h(\boldsymbol{v})\} \leq \epsilon$, then, for every unit vector $\boldsymbol{w} \in \mathbb{R}^d$ such that $\theta(\boldsymbol{v}, \boldsymbol{w}) \in [0, \pi/2]$ and $\Pr\{y = 1 \mid \mathbf{x} \in h(\boldsymbol{w})\} \geq \Omega(\epsilon^{1/4})$, there is $\langle \mathbb{E}[-g_{\boldsymbol{w}}(\mathbf{x}, y)], \bar{\boldsymbol{v}}_{\boldsymbol{w}^\perp} \rangle \geq \sqrt{\epsilon}$.*

We leave the formal proof to Appendix D due to the page limit, but sketch the proof idea as follows (also see Figure 1a). When a homogeneous halfspace $h(\boldsymbol{w})$ is substantially sub-optimal, the probability of true labels within the domain of disagreement with the optimal halfspace $h(\boldsymbol{v})$, i.e. $h(\boldsymbol{w}) \backslash h(\boldsymbol{v})$, must be large. However, the same probability cannot be too large in the optimal halfspace $h(\boldsymbol{v})$ and, hence, $h(\boldsymbol{v}) \cap h(\boldsymbol{w})$. Then, if the underlying distribution has a well-behaved $\mathbf{x}$-marginal, it implies that the $l_2$ norm of the expectation of $\mathbf{x}$ within that domain is also large.

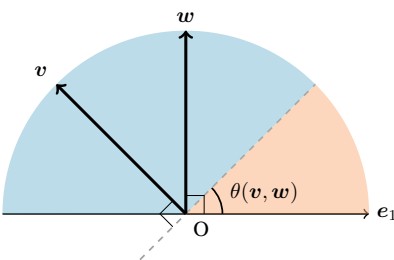 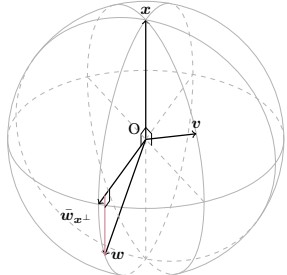

(a) blue area is $h(\boldsymbol{v}) \cap h(\boldsymbol{w})$, orange area is $h(\boldsymbol{w}) \backslash h(\boldsymbol{v})$.  (b) 3-d visualization of Contractive Projection.

Intuitively, since $\mathbb{E}[-g(\mathbf{x}, y)]$ has non-negligible projection on $\bar{\boldsymbol{v}}_{\boldsymbol{w}^\perp}$ by Lemma 3.2, it should roughly point at the same direction as the optimal normal vector $\boldsymbol{v}$ does. Hence, the gradient step (Line 4) in Algorithm 2 should move the normal vector $\boldsymbol{w}$ closer to the optimal normal vector $\boldsymbol{v}$ in each iteration. According to Diakonikolas et al. (2020a), this movement can be translated to correlation improvement, i.e., $\langle \boldsymbol{w}^{(i)}, \boldsymbol{v} \rangle > \langle \boldsymbol{w}^{(i-1)}, \boldsymbol{v} \rangle + \Omega(1)$, which, in turn, implies $\boldsymbol{w}^{(i)}$ is closer to $\boldsymbol{v}$ in terms of angle. We formally state the angle contraction guarantee in the following lemma (see Appendix D for proofs).

**Lemma 3.3** (Angle Contraction). *Fix a unit vector $\boldsymbol{v} \in \mathbb{R}^d$, $\phi \in (0, \pi/2]$, and $\kappa > 0$, let $\boldsymbol{w}, \boldsymbol{u} \in \mathbb{R}^d$ be any vectors such that $\theta(\boldsymbol{w}, \boldsymbol{v}) \in [\phi, \pi/2]$, $\langle \bar{\boldsymbol{v}}_{\boldsymbol{w}^\perp}, \boldsymbol{u} \rangle \geq \kappa$, and $\langle \boldsymbol{w}, \boldsymbol{u} \rangle = 0$. If $\boldsymbol{w}' = (\boldsymbol{w} + \lambda \boldsymbol{u}) / \|\boldsymbol{w} + \lambda \boldsymbol{u}\|_2$ with $\lambda = \kappa \phi / 4$, it holds that $\theta(\boldsymbol{w}', \boldsymbol{v}) \leq \theta(\boldsymbol{w}, \boldsymbol{v}) - \kappa^2 \phi / 64$.*

Recall that, in reference class learning, we not only wish to obtain a small $\Pr\{y = 1 \mid \mathbf{x} \in h(\boldsymbol{w})\}$, but also are required to satisfy the condition that $\boldsymbol{x} \in h(\boldsymbol{w})$. Even though Lemma 3.3 guarantees us that $\theta(\boldsymbol{u}^{(i)}, \boldsymbol{v})$ is smaller than $\theta(\boldsymbol{w}^{(i-1)}, \boldsymbol{v})$ given Lemma 3.2 holds, $\boldsymbol{u}^{(i)}$ could still "walk" out of the halfspace defined by the normal vector $\boldsymbol{x}$ or, equivalently, $\boldsymbol{x} \notin h(\boldsymbol{u}^{(i)})$. Therefore, if $\theta(\boldsymbol{u}^{(i)}, \boldsymbol{x}) \geq \pi/2$, we need to project it back onto the halfspace $h(\boldsymbol{x})$ (line 5-9) in Algorithm 2 to make sure the resulting $\boldsymbol{w}^{(i)}$ satisfies $\theta(\boldsymbol{w}^{(i)}, \boldsymbol{x}) \in [0, \pi/2]$. In fact, we can prove that such a projection is always contractive in Lemma 3.3. We defer the proof to Appendix D as it involves a lot of tedious vector decompositions, while the angle contraction can be illustrated by Figure 1b.

**Lemma 3.4** (Contractive Projection). *Fix $\boldsymbol{x}, \boldsymbol{v} \in \mathbb{R}^d$ such that $\|\boldsymbol{v}\|_2 = 1$ and $\langle \bar{\boldsymbol{x}}, \boldsymbol{v} \rangle \geq 0$. For any unit vector $\boldsymbol{w} \in \mathbb{R}^d$ that satisfies $\langle \boldsymbol{w}, \bar{\boldsymbol{x}} \rangle < 0$ and $\langle \boldsymbol{w}, \boldsymbol{v} \rangle \geq 0$, it holds that $\theta(\bar{\boldsymbol{w}}_{\boldsymbol{x}^\perp}, \boldsymbol{v}) \leq \theta(\boldsymbol{w}, \boldsymbol{v})$.*

It is clear now that, by applying Lemma 3.2 and Lemma 3.3 (and Lemma 3.4 if $\theta(\boldsymbol{u}^{(i)}, \boldsymbol{x}) \geq \pi/2$), we have that the angle between $\boldsymbol{w}$ and $\boldsymbol{v}$ will decrease by $\text{poly}(\epsilon)$ amount in each iteration until $\Pr\{y = 1 \mid \mathbf{x} \in h(\boldsymbol{w})\} = O(\epsilon^{1/4})$. Because small $\theta(\boldsymbol{w}, \boldsymbol{v})$ implies small $\Pr\{y = 1 \mid \mathbf{x} \in h(\boldsymbol{w})\}$ under well-behaved distributions, it suffices to run at most $T = 1/\text{poly}(\epsilon)$ iterations in Algorithm 2 to guarantee the existence of a good normal vector in $\mathcal{W} = \{\boldsymbol{w}^{(0)}, \ldots, \boldsymbol{w}^{(T)}\}$.

**Proposition 3.5** (Optimality Of Projected Gradient Descent). *Let $\mathcal{D}$ be any distribution on $\mathbb{R}^d \times \{0,1\}$ with centered well-behaved $\mathbf{x}$-marginal and $\boldsymbol{x} \in \mathbb{R}^d$ be an observation example. If there exists a unit vector $\boldsymbol{v} \in \mathbb{R}^d$ such that $\boldsymbol{x} \in h(\boldsymbol{v})$ and $\Pr\{y = 1 \mid \mathbf{x} \in h(\boldsymbol{v})\} \leq \epsilon$, then, Algorithm 2 runs in time at most $\tilde{O}(d\epsilon^{-9/4})$ and outputs a list $\mathcal{W}$, where there exists a $\boldsymbol{w} \in \mathcal{W}$ that satisfies both $\boldsymbol{x} \in h(\boldsymbol{w})$ and $\Pr\{y = 1 \mid \mathbf{x} \in h(\boldsymbol{w})\} \leq O(\epsilon^{1/4})$ with high probability.*

---

**Algorithm 3** Personalized Prediction

---

1: **procedure** PERPREDICT($\mathcal{D}, \text{opt}, \boldsymbol{x}, s, \epsilon, \delta$)
2:    $m \leftarrow O((s \log d + \log \frac{2}{\delta})/\epsilon^4)$
3:    $L \leftarrow$ SPARSELIST($\mathcal{D}, m, s$)
4:    $\mathcal{W} \leftarrow \{\varnothing\}$
5:    **for** $c \in L$ **do**
6:       $\mathcal{D}^{(c)} \leftarrow \mathcal{D}_\mathbf{x} \times \mathbb{1}\{c(\mathbf{x}) = y\}$
7:       $\boldsymbol{w}^{(c)} \leftarrow$ REFCLASS $\left(\mathcal{D}^{(c)}, \text{opt} + \epsilon^4, \delta/2\,|L|\,, \boldsymbol{x}\right)$
8:       $\mathcal{W} \leftarrow \mathcal{W} \cup \left\{(c, \boldsymbol{w}^{(c)})\right\}$
9:    **end for**
10:   $\hat{\mathcal{D}} \leftarrow O(\ln (d/\epsilon\delta)/\epsilon^2)$ i.i.d. samples of $\mathcal{D}$
11:   $c^*, \boldsymbol{w}^* \leftarrow \min_{\mathcal{W}} \Pr_{\hat{\mathcal{D}}} \left\{c(\mathbf{x}) \neq y \mid \mathbf{x} \in h(\boldsymbol{w}^{(c)})\right\}$
12:   **return** $c^*(\boldsymbol{x})$
13: **end procedure**

---

## 4 APPLICATION: PERSONALIZED PREDICTION

Recall that the objective of *personalized prediction* is to learn a predictor $c : \mathbb{R}^d \to \{0, 1\}$ that performs well on a given query point $\boldsymbol{x} \in \mathbb{R}^d$. As discussed previously, an intuitively good strategy to learn such a *personalized* predictor is to jointly find a pair of a classifier $c$ and a subset $S \subseteq \mathbb{R}^d$ such that not only the predictor $c$ performs well on $S$ but also the points in $S$ *resemble* $\boldsymbol{x}$.

In this section, we consider learning such a classifier-subset pair for the query point $\boldsymbol{x}$ such that $\Pr_{\mathcal{D}}\{c(\mathbf{x}) \neq y \mid \mathbf{x} \in S\}$ is minimized subject to $\boldsymbol{x} \in S$. We give a computationally efficient personalized prediction scheme for *sparse linear classifiers* and *homogeneous halfspaces* by leveraging the learning algorithm (cf. Algorithm 1) for reference classes as described in Section 3 as well as a *robust list learning* algorithm (cf. Algorithm 4) for sparse linear representations. More specifically, recall that Algorithm 1 in Section 3 guarantees to return us a homogeneous halfspace $h(\boldsymbol{w}^*) \subseteq \mathbb{R}^d$ for any given query $\boldsymbol{x} \in \mathbb{R}^d$ such that $\Pr_{(\mathbf{x},y)\sim\mathcal{D}}\{y = 1 \mid \mathbf{x} \in h(\boldsymbol{w}^*)\}$ is approximately maximized and $\boldsymbol{x} \in h(\boldsymbol{w}^*)$ over any distribution $\mathcal{D}$ with well-behaved $\mathbf{x}$-marginals. Suppose now that, for some query point $\boldsymbol{x}$, we have some binary classifier $c$ such that

$$\min_{\boldsymbol{u} \in \mathbb{R}^d : \boldsymbol{x} \in h(\boldsymbol{u})} \Pr_{(\mathbf{x},y)\sim\mathcal{D}}\{c(\mathbf{x}) = y \mid \mathbf{x} \in h(\boldsymbol{u})\} \geq 1 - \text{opt}, \tag{1}$$

we can run Algorithm 1 on the labels, $\mathbb{1}\{c(\mathbf{x}) = y\}$, with the same $\mathbf{x}$-marginal to obtain a homogeneous halfspace $h(\boldsymbol{w}^*)$ such that both $\boldsymbol{x} \in h(\boldsymbol{w}^*)$ and $\Pr\{c(\mathbf{x}) = y \mid \mathbf{x} \in h(\boldsymbol{w}^*)\} \geq 1 - O(\text{opt}^{1/4})$.

Note that, if we can find such a good classifier for the query $\boldsymbol{x}$, our algorithm for learning reference classes could approximately verify its performance on some homogeneous halfspace that contains $\boldsymbol{x}$. Therefore, the question is how to find the personalized classifier for the given query. Fortunately, a list learning algorithm for sparse linear representations can return us a small list of sparse linear classifiers, at least one of which will satisfy the optimality condition (1) (see Appendix B for details).

**Definition 4.1** (Robust list learning). *Let* $\mathcal{D} = \alpha\mathcal{D}^* + (1 - \alpha)\tilde{\mathcal{D}}$ *for an* inlier *distribution* $\mathcal{D}^*$ *and* outlier *distribution* $\tilde{\mathcal{D}}$ *each supported on* $\mathbb{R}^d \times \{0, 1\}$ *with* $\alpha \in (0, 1)$. *A robust list learning* algorithm *for a class of Boolean classifiers* $\mathcal{C}$ *will produce a finite list* $\{h_1, \ldots, h_\ell\} \subseteq \mathcal{C}$ *for some* $c^* \in \mathcal{C}$ *efficiently such that* $\max_{i=1,\ldots,l} \Pr_{\mathcal{D}^*}\{h_i(\mathbf{x}) = c^*(\mathbf{x})\} \geq 1 - \epsilon$ *with probability* $1 - \delta$.

As with Huang and Juba (2025), we obtain our main result by using the $(md)^{O(1)}$ time algorithm (with a sample of size $m$) for list learning sparse linear classifiers from a sample of size $O(\frac{1}{\alpha\epsilon}(\log d + \log \frac{1}{\delta}))$ (Juba, 2017; Mossel and Sudan, 2016). We show both theoretical analysis and experiments of our personalized prediction approach (cf. Algorithm 3) in the following sections.

### 4.1 ALGORITHM AND PERFORMANCE ANALYSIS

As an overview, Algorithm 3 first calls a robust list learning algorithm (cf. Algorithm 4) to generate a list of sparse linear classifiers $L$ (Line 2-4) and, then, calls the reference class learning algorithm for each such sparse classifier in $L$ to obtain a homogeneous halfspace (Line 5-10). At last, we sample a

small set of examples to compute the empirical risk minimizer over all the classifier-halfspace pairs. Notice that, if $L$ returned by SPARSELIST contains some classifier $c'$ that (approximately) satisfies the optimality condition (1), the optimality of Algorithm 3 follows immediately from that of Algorithm 1 (cf. Theorem 3.1) by standard concentration analysis. Therefore, the existence of an (approximately) optimal sparse classifier $c'$ in the candidate list $L$ is crucial for proving the performance guarantee of Algorithm 3, which can be formalized as the theorem below.

**Theorem 4.2** (Personalized Prediction). *Let $\mathcal{D}$ be a distribution on $\mathbb{R}^d \times \{0, 1\}$ with well-behaved $\mathbf{x}$-marginal, $\mathcal{C}$ be a class of sparse linear classifiers, and $\boldsymbol{x} \in \mathbb{R}^d$ be a query point. If there exists some $(c, \boldsymbol{v}) \in \mathcal{C} \times \mathbb{R}^d$ such that $\boldsymbol{x} \in h(\boldsymbol{v})$ and $\Pr\{c(\mathbf{x}) \neq \mathbf{y} \mid \mathbf{x} \in h(\boldsymbol{v})\} \leq \text{opt}$, then, Algorithm 3 will run in time $\text{poly}(d, 1/\epsilon, 1/\delta)$ and find some $(c^*, \boldsymbol{w}^*) \in \mathcal{C} \times \mathbb{R}^d$ such that $\boldsymbol{x} \in h(\boldsymbol{w}^*)$ and $\Pr\{c^*(\mathbf{x}) \neq \mathbf{y} \mid \mathbf{x} \in h(\boldsymbol{w}^*)\} = O(\text{opt}^{1/4}) + \epsilon \text{ w.p. } 1 - \delta$.*

We defer the proof to Appendix E. As the proof sketch, note that the sample distribution $\mathcal{D}$ can be viewed as a convex combination of a noiseless distribution $\mathcal{D}^*$, whose labels are determined by some sparse linear classifier, and a noisy distribution $\tilde{\mathcal{D}}$, whose labels are produced arbitrarily. Observe that this decomposition of $\mathcal{D}$ matches exactly with the definitions inlier and outlier distributions in the robust list learning problem (cf. Definition 4.1). As SPARSELIST (cf. Algorithm 4) is guaranteed to solve the robust list learning task with arbitrary precision (cf. Theorem B.2), at least one of the sparse classifiers in $L$ must be (approximately) optimal in the form of inequality (1).

## 4.2 EXPERIMENTS

Table 2: Test error rates. TOTAL and LIST denote the number of examples used in the entire training process (Algorithm 3 and baseline models) and the list learning (Algorithm 4) only. The models from left (LOGREG) to right (PERS) are logistic regression, SVM with Linear, RBF kernel, XGBoost tree, random forest, ERM sparse classifier (SPARSE), and personalized prediction (PERS) respectively. $^*$ indicates statistically significant improvement with 95% confidence (over SPARSE for PERS, and over PERS for the other baselines). For Pima and Hepa, PERS obtains improvement over SPARSE with 85% confidence, and the difference from the other baselines is not significant at this level.

| D/S | TOTAL | LIST | DIM | LOGREG | LIN | RBF | XGB | FOREST | SPARSE | PERS |
|---|---|---|---|---|---|---|---|---|---|---|
| HABE | 204 | 204 | 3 | .2647 | .2647 | .2941 | .3529 | .3039 | .2745 | .2745 |
| PIMA | 512 | 192 | 8 | .2461 | .25 | .2344 | .2344 | .2304 | .2852 | .2461 |
| HEPA | 103 | 103 | 20 | .1538 | .1538 | .1346 | .2115 | .1538 | .2308 | .1538 |
| HYPO | 2109 | 64 | 20 | .0199$^*$ | .019$^*$ | .0285 | .0133$^*$ | .0142$^*$ | .0579 | .0379$^*$ |
| WDBC | 379 | 48 | 30 | .0368 | .0474 | .0421 | .0421 | .0579 | .0789 | .0474$^*$ |

We evaluated our algorithms on several UCI medical datasets that are commonly used as benchmarks (Grandvalet et al., 2008; Wiener and El-Yaniv, 2011; 2015). We compare our result to a few standard machine learning models as shown in Table 2. We stress that our method differs from these standard models in the key respect that we obtain a 2-sparse linear classifier whose decision making is inherently interpretable, whereas the other models are typically not humanly understandable. More detailed analysis will be presented in Appendix F due to page limitation.

## 5 LIMITATIONS AND FUTURE DIRECTIONS

Several questions naturally present themselves for future work. The first question is whether our $O(\text{opt}^{1/4})$ error bound can be improved for a similarly broad family of distributions, perhaps by assuming some additional (natural) properties. The second is how we might target different coverage levels. Although Huang and Juba (2025) obtained a $1/\sqrt{\log d}$ additive lower bound, obtaining a multiplicative upper/lower bound for general halfspaces is still an open question, even for Gaussian marginals. Also, alternatively, we could consider families of non-homogeneous halfspaces that are still not completely general, such as halfspaces with bounded coefficients. And, finally, we were restricted to the use of sparse linear classifiers because this was the only family of classifiers for which we had a strong robust learning guarantee. It would be interesting to learn other classes, perhaps using similar kinds of distributional assumptions.

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

## A    OMITTED DEFINITIONS AND PROOFS IN SECTION 2

For completeness, we give a formal definition of *conditional classification* problem following Huang and Juba (2025).

**Definition A.1** (Conditional Classification). *Let $\mathcal{D}$ be any probability distribution over $\mathbb{R}^d \times \{0, 1\}$, $\mathcal{C} \subseteq \{c : \mathbb{R}^d \to \{0, 1\}\}$ be a class of classifiers, and $\mathcal{H}$ be a collection of subsets of $\mathbb{R}^d$. For parameters $\alpha > 0$ and $\epsilon, \delta \in (0, 1)$, the $\alpha$-approximate Conditional Classification problem is, given $m$ labeled examples drawn from $\mathcal{D}$, to return a pair $(c, S) \in \mathcal{C} \times \mathcal{H}$ such that with probability $1 - \delta$, for any $(c^*, S^*) \in \mathcal{C} \times \mathcal{H}$,*

$$\Pr_{(\mathbf{x},\mathbf{y}) \sim \mathcal{D}}\{c(\mathbf{x}) \neq \mathbf{y} \mid \mathbf{x} \in S\} \leq \alpha \Pr_{(\mathbf{x},\mathbf{y}) \sim \mathcal{D}}\{c^*(\mathbf{x}) \neq \mathbf{y} \mid \mathbf{x} \in S^*\} + \epsilon.$$

*If $\alpha = 1$, we simply refer to the problem as Conditional Classification.*

Now we prove that the *personalized prediction* problem is at least as hard as conditional classification.

**Claim A.2** (Claim 2.3). *There is an efficient reduction from conditional classification to personalized prediction whenever there is a population lower bound on the subset class.*

*Proof.* With a population lower bound $\mu \in (0, 1)$, we may obtain an example inside the optimal subset of the conditional classification instance with high probability by sampling $O(1/\mu)$ points. By using these points as the observations and taking the best reference class as our output, solving the personalized prediction problem for the same hypothesis classes enables us to efficiently solve the conditional classification instance. □

## B    REVIEW OF ROBUST LIST LEARNING OF SPARSE LINEAR CLASSIFIERS

For completeness, we give the formal definition of Robust List Learning problem as follow:

**Definition B.1** (Definition 4.1). *Let $\mathcal{D} = \alpha \mathcal{D}^* + (1 - \alpha)\tilde{\mathcal{D}}$ for an inlier distribution $\mathcal{D}^*$ and outlier distribution $\tilde{\mathcal{D}}$ each supported on $\mathbb{R}^d \times \{0, 1\}$, with $\alpha \in (0, 1)$. A robust list learning algorithm for a class of Boolean classifiers $\mathcal{C}$, given $\alpha$ and parameters $\epsilon, \delta \in (0, 1)$, and sample access to $\mathcal{D}$ such that for $(\mathbf{x}, \mathbf{b})$ in the support of $\mathcal{D}^*$, $\mathbf{b} = c^*(\mathbf{x})$ for some $c^* \in \mathcal{C}$, runs in time $\mathrm{poly}(d, \frac{1}{\alpha}, \frac{1}{\epsilon}, \log \frac{1}{\delta})$, and with probability $1 - \delta$ returns a list of $\ell = \mathrm{poly}(d, \frac{1}{\alpha}, \frac{1}{\epsilon}, \log \frac{1}{\delta})$ classifiers $\{h_1, \ldots, h_\ell\}$ such that for some $h_i$ in the list, $\Pr_{\mathcal{D}^*}\{h_i(\mathbf{x}) = c^*(\mathbf{x})\} \geq 1 - \epsilon$.*

---

**Algorithm 4** Robust list learning of sparse linear classifiers

1:  **procedure** SPARSELIST($\mathcal{D}, m, s$)
2:      $L \leftarrow \varnothing$
3:      $\nu \leftarrow 2^{-(bs + s \log s)}$
4:      Sample $(\mathbf{x}^{(1)}, \mathbf{y}^{(1)}), \ldots, (\mathbf{x}^{(m)}, \mathbf{y}^{(m)}) \sim \mathcal{D}$
5:      Re-map $\mathbf{y}^{(i)}$ from $\{0, 1\}$ to $\{-1, +1\}$ for all $i \in [m]$
6:      **for** $(i_1, \ldots, i_s) \in [d]^s$ and $(j_1, \ldots, j_s) \in [m]^s$ **do**

7:      $\boldsymbol{w} \leftarrow \begin{bmatrix} \mathbf{y}^{(j_1)}\mathbf{x}_{i_1}^{(j_1)} & \cdots & \mathbf{y}^{(j_1)}\mathbf{x}_{i_s}^{(j_1)} \\ & \vdots & \\ \mathbf{y}^{(j_s)}\mathbf{x}_{i_1}^{(j_s)} & \cdots & \mathbf{y}^{(j_s)}\mathbf{x}_{i_s}^{(j_s)} \end{bmatrix}^{-1} \begin{bmatrix} \mathbf{y}^{(j_1)} - \nu \\ \vdots \\ \mathbf{y}^{(j_s)} - \nu \end{bmatrix}$

8:      $L \leftarrow L \cup \{\boldsymbol{w}\}$
9:      **end for**
10:      **return** $L$
11:  **end procedure**

---

For completeness, we now describe an algorithm to solve the robust list learning problem for sparse linear classifiers. It is based on the approach used in the algorithm for conditional sparse linear regression Juba (2017), using an observation by Mossel and Sudan (2016). We will prove the following:

**Theorem B.2** (Mossel and Sudan (2016); Juba (2017); Huang and Juba (2025)). *Suppose the numbers are $b$-bit rational values, Algorithm 4 solves robust list-learning of linear classifiers with $s = O(1)$ nonzero coefficients, margin $\nu \geq 2^{-(bs+s\log s)}$, and probability at least $1 - \delta$ from $m = O(\frac{1}{\alpha\epsilon}(s\log d + \log\frac{1}{\delta}))$ examples in polynomial time with list size $O((md)^s)$.*

*Proof.* We observe that the running time and list size of Algorithm 4 is clearly as promised. To see that it solves the problem, we first recall that results by Blumer et al. (1989) and Hanneke (2016) showed that given $O(\frac{1}{\epsilon}(D + \log\frac{1}{\delta}))$ examples labeled by a class of VC-dimension $D$, any consistent hypotheses achieves error $\epsilon$ with probability $1 - \delta$. In particular, halfspaces in $\mathbb{R}^d$ have VC-dimension $d$; Haussler (1988) observed that $s$-sparse linear classifiers in $\mathbb{R}^d$ have VC-dimension $s\log d$. Hence, if the data includes a set $S$ of at least $\Omega(\frac{1}{\epsilon}(s\log d + \log\frac{1}{\delta}))$ inliers and we find a $s$-sparse classifier that agrees with the labels on $S$, it achieves error $1 - \epsilon$ on $S$ with probability $1 - \delta/2$. Observe that in a sample of size $O(\frac{1}{\alpha\epsilon}(s\log d + \log\frac{1}{\delta}))$, with an $\alpha$ fraction of inliers, we indeed obtain $\Omega(\frac{1}{\epsilon}(s\log d + \log\frac{1}{\delta}))$ inliers with probability $1 - \delta/2$.

Now, suppose we write our linear threshold function with a standard threshold of 1, and suppose are examples are drawn from $\mathbb{R}^d \times \{-1, 1\}$. Then a classifier with weight vector $\boldsymbol{w}$ labels $\mathbf{x}$ with 1 if $\langle \boldsymbol{w}, \mathbf{x} \rangle \geq 1$, and labels $\mathbf{x}$ with $-1$ if $\langle \boldsymbol{w}, \mathbf{x} \rangle < 1$. We observe that by Cramer's rule, we can find a value $\nu^* > 0$ (of size at least $2^{-(bs+s\log s)}$ if the numbers are $b$-bit rational values) such that if $\langle \boldsymbol{w}, \mathbf{x} \rangle < 1$, $\langle \boldsymbol{w}, \mathbf{x} \rangle \leq 1 - \nu^*$. So, it is sufficient for $\langle \boldsymbol{w}, \mathrm{y}\mathbf{x} \rangle \geq \mathrm{y} - \nu$ for a given $(\mathbf{x}, \mathrm{y})$, for some margin $\nu \geq 2^{-(bs+s\log s)}$. Thus, to find a consistent $\boldsymbol{w}$, it suffices to solve the linear program $\langle \boldsymbol{w}, \mathrm{y}^{(j)}\mathbf{x}^{(j)} \rangle \geq \mathrm{y}^{(j)} - \nu$ for each $j$th example in $S$. Observe that if we parameterize $\boldsymbol{w}$ by only the nonzero coefficients, we obtain a linear program in $s$ dimensions, for which we can obtain a feasible solution at a vertex, given by $s$ tight constraints. Now, Algorithm 4 enumerates *all* $s$-tuples of indices and examples, which in particular therefore must include any $s$-tuple of examples in the inlier set $S$ and the $s$ nonzero coordinates of $\boldsymbol{w}$. Hence, with probability at least $1 - \delta$, $L$ indeed contains some $\boldsymbol{w}$ that attains error $\epsilon$ on $S$, as needed. $\qquad\square$

## C   WELL-BEHAVED DISTRIBUTIONS

We recall the formal definition of the family of **well-behaved** distributions as follows:

**Definition C.1** (Well-Behaved Distributions). *A distribution $\mathcal{D}_\mathbf{x}$ on $\mathbb{R}^d$ is said to be $(K, U, L, R)$-well-behaved if the following properties hold:*

1. *$K$-**bounded**: there exists a constant $K$ such that $\hat{\|}\langle \mathbf{x}, \boldsymbol{u} \rangle \hat{\|}_p \leq Kp$ for all unit vectors $\boldsymbol{u} \in \mathbb{R}^d$ and $p \geq 1$.*

2. *$U$-**concentration and anti-concentration**: let $V$ be any subspace with dimensionality at most 2 and $\gamma_V$ be the corresponding probability density function of $\mathcal{D}_\mathbf{x}$ on $\mathbb{R}^2$ when projected onto $V$. Then, for all $\mathbf{x} \in V$, there exists a non-negative function $p : \mathbb{R}_+ \to \mathbb{R}_+$ such that $\gamma_V(\mathbf{x}) \leq p(\|\mathbf{x}\|_2) \leq U$ and $\int_V \|\mathbf{x}\|_2 p(\|\mathbf{x}\|_2)d\mathbf{x} \leq U$.*

3. *$L$-**anti-anti-concentration**: let $\gamma_{\boldsymbol{u}}$ be the marginal density function of $\langle \mathbf{x}, \boldsymbol{u} \rangle$ for any unit vector $\boldsymbol{u} \in \mathbb{R}^d$. Then $\gamma_{\boldsymbol{u}}(\langle \mathbf{x}, \boldsymbol{u} \rangle) \geq L$ for all $|\langle \mathbf{x}, \boldsymbol{u} \rangle| \leq 1$.*

4. *$R$-**rounded**: $\mathrm{Pr}_{\mathbf{x} \in \mathcal{D}_\mathbf{x}}\{\mathbf{x} \in h_t(\boldsymbol{u})\} \geq R$ for all halfspaces $h_t(\boldsymbol{u}) \subseteq \mathbb{R}^d$ such that $\mathbb{E}_{\mathbf{x} \sim \mathcal{D}_\mathbf{x}}[\mathbf{x}] \in h_t(\boldsymbol{u})$.*

In comparison to the class of distributions considered by Diakonikolas et al. (2020c) for agnostic classification, we require two additional properties, boundedness and roundedness. Notice that the $K$-bounded property is equivalent to a sub-exponential tail bound Vershynin (2018). Roundedness can be ensured in polynomial time by centering the data Har-Peled and Jones (2021), though this of course changes the sets corresponding to homogeneous halfspaces. One can verify that the distributions satisfying our definition include a wide variety of classes such as log-concave distributions Lovász and Vempala (2007).

Let's see a few specific examples of well-behaved distributions.

**Example C.2** (Gaussian Distribution). *Any Gaussian distribution $\mathcal{N}^d(0, \sigma^2)$ is a well-behaved distribution with $K = \sigma$, $U = \max((\sigma\sqrt{2\pi})^{-3/2}, \sqrt{3} + O(\sigma^2))$, $L = e^{\sigma^{-2}/2}/\sigma\sqrt{2\pi}$, and $R = 1/2$.*

*Proof.* Let's first notice that the projection of a random vector $\mathbf{x} \sim \mathcal{N}^d(0, \sigma^2)$ onto a $k \leq d$ dimension subspace will result to $\mathbf{z} \sim \mathcal{N}^k(0, \sigma^2)$.

To show $K = \sigma$, by the *integral identity*, we have that

$$\hat{\|}\langle \mathbf{x}, \boldsymbol{u}\rangle\|_p^p = \int_0^\infty \Pr_{z \sim \mathcal{N}(0,\sigma^2)}\{z^p \geq u\}du$$

$$\overset{(i)}{=} \int_0^\infty \Pr_{z \sim \mathcal{N}(0,\sigma^2)}\{|z| \geq t\}pt^{p-1}dt$$

$$\overset{(ii)}{\leq} \int_0^\infty 2e^{-t^2/2\sigma^2}pt^{p-1}dt$$

$$\overset{(iii)}{=} \left(\sigma\sqrt{2}\right)^p p\Gamma(p/2)$$

$$\leq \left(\sigma\sqrt{2}\right)^p p\left(p/2\right)^{p/2}$$

where inequality (i) is obtained by change of variables $u = t^p$. Inequality (ii) holds due to Fact G.1. Then, setting $t^2 = 2\sigma^2 s$ and using definition of Gamma function give inequality (iii). And the last inequality holds since $\Gamma(x) \leq x^x$ by Stirling's approximation. Taking the $p$th root over the above inequality gives the first property.

For the second property $U = \max((\sigma\sqrt{2\pi})^{-3/2}, \sqrt{3} + O(\sigma^2))$, notice that the density of any $k$-dimensional $0$-mean Gaussian distribution is upper bounded by $\left(\sigma\sqrt{2\pi}\right)^{-k/2}$ by definition. Meanwhile, taking $p$ to be the density of such Gaussian distribution, it holds that

$$\int_{\mathbb{R}^k} \|\mathbf{z}\|_2 p(\|\mathbf{z}\|_2)d\mathbf{z} = \int_{\mathbb{R}^k} \|\mathbf{z}\|_2 \phi(\|\mathbf{z}\|_2)d\mathbf{z}$$

$$= \mathbb{E}_{\mathbf{z} \sim \mathcal{N}^k(0,\sigma^2)}[\|\mathbf{z}\|_2]$$

$$\leq \sqrt{k} + O(\sigma^2)$$

where the last inequality can be acquired by referring to Exercise 3.1.4. of Vershynin (2018). This implies the claimed property.

The third property $L = e^{\sigma^{-2}/2}/\sigma\sqrt{2\pi}$ holds because the density function of a one dimension Gaussian distribution is monotonically decrease from $0$ to $1$.

The last property is obvious. $\qquad\square$

To see another example, we first define the $d$-dimensional hyper-ball as follows.

**Definition C.3** ($d$-Dimensional Hyper-Ball). *For any $r > 0$ and $\boldsymbol{\mu} \in \mathbb{R}^d$, we define*

$$\mathcal{B}^d(\boldsymbol{\mu}, r) = \left\{\mathbf{x} \in \mathbb{R}^d \mid \|\mathbf{x} - \boldsymbol{\mu}\|_2 \leq r\right\}$$

*to be the $d$-dimensional hyper-ball of radius $r$ centered at $\boldsymbol{\mu}$.*

**Fact C.4** (Volume Of Hyper-Ball). *There is $\mathrm{Vol}(\mathcal{B}^d(0, r)) = \pi^{d/2}r^d/\Gamma(d/2 - 1)$.*

Now, we show that the uniform distribution over a large variety of compact sets are also well-behaved.

**Example C.5** (Uniform Distribution Over Compact Sets). *Let $\mathrm{Unif}(S)$ denote the uniform distribution over any $S \subseteq \mathbb{R}^d$, and $T \subset \mathbb{R}^d$ be a compact set such that $\mathrm{Vol}(T) = \nu$, $\max_{\mathbf{x} \in T}\|\mathbf{x}\|_2 \leq \tau$ for some $\tau \geq 1$, and $\sup\left\{r \mid \mathcal{B}^d(\boldsymbol{\mu}_T, r) \subseteq T\right\} \geq 1$ where $\boldsymbol{\mu}_T = \mathbb{E}_{\mathbf{x} \sim \mathrm{Unif}(T)}[\mathbf{x}]$. Then, $\mathrm{Unif}(T - \boldsymbol{\mu}_T)$ is a well-behaved distribution such that*

$$K = \tau, \quad U \approx \max\left(\frac{\tau^{d'}}{\nu\sqrt{d'\pi}}\left(\frac{2\pi e}{d'}\right)^{d'/2}, \tau\right), \quad L \approx \frac{1}{\nu\sqrt{d'\pi}}\left(\frac{2\pi e}{d'}\right)^{d'/2}, \quad R \approx \frac{1}{2\nu\sqrt{d'\pi}}\left(\frac{2\pi e}{d'}\right)^{d'/2}$$

*where $d' = d - \dim(V)$ for any subspace $V$ of dimension at most $3$.*

*Proof.* To show the $K$-boundedness, let's first notice that $\langle \mathbf{x} - \boldsymbol{\mu}_T, \boldsymbol{u} \rangle \leq \|\mathbf{x} - \boldsymbol{\mu}_T\|_2$ by the Cauchy-Schwartz inequality. Then, similar to the Gaussian example, we have that

$$\hat{\|}\langle \mathbf{x} - \boldsymbol{\mu}_T, \boldsymbol{u} \rangle\hat{\|}_p^p = \int_0^\infty \mathrm{Pr}_{\mathbf{x}\sim\mathrm{Unif}(T)}\{\|\mathbf{x} - \boldsymbol{\mu}_T\|_2^p \geq u\}du$$

$$\overset{\text{(i)}}{=} \int_0^{\tau^p} \mathrm{Pr}_{\mathbf{x}\sim\mathrm{Unif}(T)}\{\|\mathbf{x} - \boldsymbol{\mu}_T\|_2^p \geq u\}du$$

$$\overset{\text{(ii)}}{\leq} \int_0^{\tau^p} 1du$$

$$\leq \tau^p$$

where inequality (i) holds because $\max_{\mathbf{x}\in T}\|\mathbf{x}\|_2 \leq \tau$ and inequality (ii) holds because any probability is less than or equal to 1. Again, take the $p$th root over the above inequality gives the first property.

For the second property, denote $\mathbf{z} = \mathbf{x} - \boldsymbol{\mu}_T$, $d' = d - \dim(V)$, and $\mathrm{proj}_V(S) = \{\mathbf{x}_V \mid \mathbf{x} \in S\}$, we have that

$$\gamma_V(\mathbf{z}) = \int_{\mathrm{proj}_{V^\perp}(T)} \frac{1}{\nu}d\mathbf{z}$$

$$\overset{\text{(i)}}{\leq} \frac{1}{\nu} \int_{\mathrm{proj}_{V^\perp}(\mathcal{B}^d(0,\tau))} d\mathbf{z}$$

$$\overset{\text{(ii)}}{=} \frac{1}{\nu} \int_{\mathcal{B}^{d'}(0,\tau)} d\mathbf{z}$$

$$= \mathrm{Vol}(\mathcal{B}^{d'}(0,\tau))/\nu$$

$$\overset{\text{(iii)}}{=} \frac{\pi^{d'/2}\tau^{d'}}{\nu\Gamma(d'/2 - 1)}$$

$$\approx \frac{\tau^{d'}}{\nu\sqrt{d'\pi}}\left(\frac{2\pi e}{d'}\right)^{d'/2}$$

where inequality (i) holds because $T - \boldsymbol{\mu}_T \subseteq \mathcal{B}^d(0,\tau)$. Equation (ii) holds because $V^\perp$ has dimension $d - \dim(V)$. Equation (iii) is obtained by invoking Fact C.4. The last equation results from Stirling's approximation. Meanwhile, we have that

$$\int_{\mathrm{proj}_{V^\perp}(T)} \|\mathbf{z}\|_2\gamma_V(\mathbf{z})d\mathbf{z} \leq \tau \int_{\mathrm{proj}_{V^\perp}(T)} \gamma_V(\mathbf{z})d\mathbf{z}$$

$$= \tau$$

which completes the proof for the second property.

For the third property, notice that it suffices to show this property holds for all $\|\mathbf{z}\|_2 \leq 1$. Therefore, for $\|\mathbf{z}\|_2 \leq 1$, we have that

$$\gamma_V(\mathbf{z}) = \int_{\mathrm{proj}_{V^\perp}(T)} \frac{1}{\nu}d\mathbf{z}$$

$$\overset{\text{(i)}}{\geq} \frac{1}{\nu} \int_{\mathrm{proj}_{V^\perp}(\mathcal{B}^d(0,1))} d\mathbf{z}$$

$$= \frac{1}{\nu} \int_{\mathcal{B}^{d'}(0,\tau)} d\mathbf{z}$$

$$\overset{\text{(ii)}}{=} \frac{\pi^{d'/2}}{\nu\Gamma(d'/2 - 1)}$$

$$\approx \frac{1}{\nu\sqrt{d'\pi}}\left(\frac{2\pi e}{d'}\right)^{d'/2}$$

where inequality (i) holds because we assumed $\mathcal{B}^d(\boldsymbol{\mu}_T, 1) \subseteq T$. Inequality (ii) and the last equation hold due to, again, Fact C.4 and Stirling's approximation.

The last property holds because any halfspace containing $\boldsymbol{\mu}_T$ must also contain at least a half of the hyper-ball $\mathcal{B}^d(\boldsymbol{\mu}_T, 1)$, which has volume at least

$$\frac{1}{2\nu\sqrt{d'\pi}}\left(\frac{2\pi e}{d'}\right)^{d'/2}$$

by Fact C.4 and Stirling's approximation. $\qquad\square$

## D  ANALYSIS OF ALGORITHM 1

**Lemma D.1** (Lemma 3.2). *Let $\mathcal{D}$ be any distribution on $\mathbb{R}^d \times \{0,1\}$ with centered and $(K, U, L, R)$-well-behaved $\mathbf{x}$-marginal, and define $g_{\boldsymbol{w}}(\mathbf{x}, \mathrm{y}) = \mathrm{y} \cdot \mathbf{x}_{\boldsymbol{w}\perp}\mathbb{1}\{\mathbf{x} \in h(\boldsymbol{w})\}$. Suppose there exists a unit vector $\boldsymbol{v} \in \mathbb{R}^d$ that satisfies $\Pr_{(\mathbf{x},\mathrm{y})\sim\mathcal{D}}\{\mathrm{y} = 1 \mid \mathbf{x} \in h(\boldsymbol{v})\} \leq \epsilon$ for some sufficiently small $\epsilon \in (0, 1/2)$, then, for every unit vector $\boldsymbol{w} \in \mathbb{R}^d$ such that $\theta(\boldsymbol{v}, \boldsymbol{w}) \in [0, \pi/2]$ and*

$$\Pr_{(\mathbf{x},\mathrm{y})\sim\mathcal{D}}\{\mathrm{y} = 1 \mid \mathbf{x} \in h(\boldsymbol{w})\} \geq (U\sqrt{2(2K+1)/R^2 L} + 1/R)\epsilon^{1/4},$$

*there is*

$$\left\langle \mathbb{E}_{(\mathbf{x},\mathrm{y})\sim\mathcal{D}}[-g_{\boldsymbol{w}}(\mathbf{x}, \mathrm{y})], \bar{\boldsymbol{v}}_{\boldsymbol{w}\perp} \right\rangle \geq \sqrt{\epsilon}.$$

*Proof.* For conciseness, let $\theta = \theta(\boldsymbol{v}, \boldsymbol{w})$ and define two orthonormal basis $\boldsymbol{e}_1, \boldsymbol{e}_2$ such that $\boldsymbol{w} = \boldsymbol{e}_2$ and $\boldsymbol{v} = -\boldsymbol{e}_1 \sin\theta + \boldsymbol{e}_2 \cos\theta$, which implies $\boldsymbol{e}_1 = -\bar{\boldsymbol{v}}_{\boldsymbol{w}\perp}$. Denote $\mathrm{x}_i = \langle \mathbf{x}, \boldsymbol{e}_i \rangle$ so that $\langle \mathbf{x}, \boldsymbol{w} \rangle = \mathrm{x}_2$ and $\langle \mathbf{x}, \boldsymbol{v} \rangle = -\mathrm{x}_1 \sin\theta + \mathrm{x}_2 \cos\theta$. Because $\langle \mathbf{x}, \boldsymbol{e}_1 \rangle = \langle \mathrm{x}_2\boldsymbol{e}_2 + \mathbf{x}_{\boldsymbol{e}_2^\perp}, \boldsymbol{e}_1 \rangle = -\langle \mathbf{x}_{\boldsymbol{w}\perp}, \bar{\boldsymbol{v}}_{\boldsymbol{w}\perp} \rangle$, we have

$$\langle \mathbb{E}[-g_{\boldsymbol{w}}(\mathbf{x}, \mathrm{y})], \bar{\boldsymbol{v}}_{\boldsymbol{w}\perp} \rangle = \mathbb{E}[-\mathrm{y} \cdot \langle \mathbf{x}_{\boldsymbol{w}\perp}, \bar{\boldsymbol{v}}_{\boldsymbol{w}\perp} \rangle \mathbb{1}\{\mathbf{x} \in h(\boldsymbol{w})\}]$$

$$\overset{(i)}{=} \mathbb{E}[\mathrm{y} \cdot \langle \mathbf{x}_{\boldsymbol{w}\perp}, \boldsymbol{e}_1 \rangle \mathbb{1}\{\mathrm{x}_2 \geq 0\}]$$

$$= \mathbb{E}[\mathrm{y} \cdot \mathrm{x}_1 \mathbb{1}\{\mathrm{x}_2 \geq 0, \mathbf{x} \in h^c(\boldsymbol{v})\}] - \mathbb{E}[\mathrm{y} \cdot \mathrm{x}_1 \mathbb{1}\{\mathrm{x}_2 \geq 0, \mathbf{x} \in h(\boldsymbol{v})\}]$$

$$\geq \mathbb{E}[\mathrm{y} \cdot \mathrm{x}_1 \mathbb{1}\{\mathrm{x}_2 \geq 0, \mathbf{x} \in h^c(\boldsymbol{v})\}] - \mathbb{E}[|\mathrm{x}_1| \mathbb{1}\{\mathrm{x}_2 \geq 0, \mathbf{x} \in h(\boldsymbol{v}), \mathrm{y} = 1\}]$$

$$\overset{(ii)}{\geq} \mathbb{E}[\mathrm{y} \cdot \mathrm{x}_1 \mathbb{1}\{\mathrm{x}_2 \geq 0, \mathbf{x} \in h^c(\boldsymbol{v})\}] - \sqrt{\mathbb{E}[\mathrm{x}_1^2]\Pr\{\mathrm{x}_2 \geq t \cap \mathbf{x} \in h(\boldsymbol{v}) \cap \mathrm{y} = 1\}}$$

$$\overset{(iii)}{\geq} \mathbb{E}[\mathrm{y} \cdot \mathrm{x}_1 \mathbb{1}\{\mathrm{x}_2 \geq 0, \mathbf{x} \in h^c(\boldsymbol{v})\}] - 2K\sqrt{\Pr\{\mathrm{y} = 1 \mid \mathbf{x} \in h(\boldsymbol{v})\}\Pr\{\mathbf{x} \in h(\boldsymbol{v})\}}$$

$$\geq \underbrace{\mathbb{E}[|\mathrm{x}_1| \cdot \mathbb{1}\{\mathrm{x}_1 \tan\theta > \mathrm{x}_2 \geq 0, \mathrm{y} = 1\}]}_{I} - 2K\sqrt{\epsilon}. \qquad (2)$$

where equation (i) holds because $\mathbf{x} \in h(\boldsymbol{w})$ is equivalent to $\langle \mathbf{x}, \boldsymbol{w} \rangle \geq 0$, which is equivalent to $\mathrm{x}_2 \geq 0$ by definition, inequality (ii) holds by applying Cauchy's inequality to the second expectation, inequality (iii) is obtained since $\Pr\{\mathrm{x}_2 \geq t \cap \mathbf{x} \in h^c(\boldsymbol{v}) \cap \mathrm{y} = 1\} \leq \Pr\{\mathbf{x} \in h^c(\boldsymbol{v}) \cap \mathrm{y} = 1\}$ as well as $\mathcal{D}_{\mathbf{x}}$ is $K$-bounded, and the last inequality holds due to optimality assumption and the fact that $\Pr\{\mathbf{x} \in h(\boldsymbol{v})\} \leq 1$.

Then, we will apply lemma D.2 to lower bound $I$. Observe that the event $\mathrm{x}_1 \tan\theta > \mathrm{x}_2 \geq 0$ in $I$ is a subset of event $\mathrm{x}_1 \geq 0$ because $\theta(\boldsymbol{v}, \boldsymbol{w}) \in [0, \pi/2]$. Therefore, we can view the event $\mathrm{x}_1 \geq 0$ as $T$ in lemma D.2 and show that, by the anti-concentration property of $\mathcal{D}_{\mathbf{x}}$, there exists a $\beta > 0$ such that $\Pr\{0 \leq \mathrm{x}_1 \leq \beta\} \leq \Pr\{\mathrm{x}_1 \tan\theta > \mathrm{x}_2 \geq 0 \cap \mathrm{y} = 1\}$ to apply lemma D.2.

Observe that, due to the anti-concentration property of $\mathcal{D}_{\mathbf{x}}$, the density of $\mathrm{x}_1$ is upper bounded by $U$. Therefore, taking $\beta = \sqrt{2(2K+1)/L}\epsilon^{1/4}$ yields

$$\Pr\{0 \leq \mathrm{x}_1 \leq \beta\} \leq U\sqrt{2(2K+1)/L}\epsilon^{1/4}$$

$$= (U\sqrt{2(2K+1)/R^2 L} + \Pr\{\mathbf{x} \in h(\boldsymbol{v})\}/R)R\epsilon^{1/4} - \Pr\{\mathbf{x} \in h(\boldsymbol{v})\}\epsilon^{1/4}$$

$$\overset{(i)}{\leq} (U\sqrt{2(2K+1)/R^2 L} + 1/R)R\epsilon^{1/4} - \Pr\{\mathbf{x} \in h(\boldsymbol{w}) \cap \mathbf{x} \in h(\boldsymbol{v}) \cap \mathrm{y} = 1\}$$

$$\overset{(ii)}{\leq} R \cdot \Pr\{\mathrm{y} = 1 \mid \mathbf{x} \in h(\boldsymbol{w})\} - \Pr\{\mathbf{x} \in h(\boldsymbol{w}) \cap \mathbf{x} \in h(\boldsymbol{v}) \cap \mathrm{y} = 1\}$$

$$\overset{(iii)}{\leq} \Pr\{\mathbf{x} \in h^c(\boldsymbol{v}) \cap \mathbf{x} \in h(\boldsymbol{w}) \cap \mathrm{y} = 1\}$$

$$= \Pr\{\mathrm{x}_1 \tan\theta > \mathrm{x}_2 \geq 0 \cap \mathrm{y} = 1\}$$

where inequality (i) holds due to our assumption that $\Pr\{y = 1 \mid \mathbf{x} \in h(\boldsymbol{v})\} \le \epsilon \le \epsilon^{1/4}$ as well as the fact that $\Pr\{\mathbf{x} \in h(\boldsymbol{v})\} \le 1$, and inequality (ii) holds because we assumed $\Pr\{y = 1 \mid \mathbf{x} \in h(\boldsymbol{w})\} \ge (U\sqrt{2(2K+1)/R^2 L} + 1/R)\epsilon^{1/4}$, inequality (iii) is obtained since $\mathcal{D}_\mathbf{x}$ is $R$-rounded and centered so that $\Pr\{\mathbf{x} \in h(\boldsymbol{w})\} \ge R$.

Now, applying lemma D.2 gives

$$I \ge \mathbb{E}[x_1 \cdot \mathbb{1}\{0 \le x_1 \le \sqrt{2(2K+1)/L}\epsilon^{1/4}\}]$$
$$\overset{(i)}{\ge} L \int_0^{\sqrt{2(2K+1)/L}\epsilon^{1/4}} x_1 dx_1$$
$$= (2K+1)\sqrt{\epsilon} \tag{3}$$

where inequality (i) is due to $L$-anti-anti-concentration property of $\mathcal{D}_\mathbf{x}$. At last, taking inequality (3) back to equation (2) leads to the claimed result. $\square$

The following lemma plays a key role in proving the above proposition.

**Lemma D.2** (Lemma C.3 in Huang and Juba (2025)). *Let $\mathcal{D}$ be an arbitrary distribution on $\mathbb{R}^d$, and $S, T$ be any events such that $\Pr_\mathcal{D}\{S \cap T\} = p$ for some $p \in (0, 1)$. Then, for any unit vector $\boldsymbol{u} \in \mathbb{R}^d$, and parameters $\alpha, \beta$ that satisfies $\Pr\{T \cap |\langle \mathbf{x}, \boldsymbol{u} \rangle| \le \beta\} \le p \le \Pr\{T \cap |\langle \mathbf{x}, \boldsymbol{u} \rangle| \ge \alpha\}$, there are*

$$\mathbb{E}_\mathcal{D}[|\langle \mathbf{x}, \boldsymbol{u} \rangle| \cdot \mathbb{1}\{T, |\langle \mathbf{x}, \boldsymbol{u} \rangle| \le \beta\}] \le \mathbb{E}_\mathcal{D}[|\langle \mathbf{x}, \boldsymbol{u} \rangle| \cdot \mathbb{1}\{S, T\}] \le \mathbb{E}_\mathcal{D}[|\langle \mathbf{x}, \boldsymbol{u} \rangle| \cdot \mathbb{1}\{T, |\langle \mathbf{x}, \boldsymbol{u} \rangle| \ge \alpha\}].$$

**Lemma D.3** (Lemma 3.3). *Fix a unit vector $\boldsymbol{v} \in \mathbb{R}^d$, $\phi \in (0, \pi/2]$, and $\kappa > 0$, let $\boldsymbol{w}, \boldsymbol{u} \in \mathbb{R}^d$ be any vectors such that $\theta(\boldsymbol{w}, \boldsymbol{v}) \in [\phi, \pi/2]$, $\langle \bar{\boldsymbol{v}}_{\boldsymbol{w}^\perp}, \boldsymbol{u} \rangle \ge \kappa$, and $\langle \boldsymbol{w}, \boldsymbol{u} \rangle = 0$. If*

$$\boldsymbol{w}' = \frac{\boldsymbol{w} + \lambda \boldsymbol{u}}{\|\boldsymbol{w} + \lambda \boldsymbol{u}\|_2}$$

*with $\lambda = \kappa\phi/4$, it holds that $\theta(\boldsymbol{w}', \boldsymbol{v}) \le \theta(\boldsymbol{w}, \boldsymbol{v}) - \kappa^2\phi/64$.*

*Proof.* By the assumptions that $\langle \boldsymbol{w}, \boldsymbol{u} \rangle = 0$ and $\langle \bar{\boldsymbol{v}}_{\boldsymbol{w}^\perp}, \boldsymbol{u} \rangle \ge \kappa$, we must have that

$$\langle \boldsymbol{v}, \boldsymbol{u} \rangle = \|\boldsymbol{v}_{\boldsymbol{w}^\perp}\|_2 \langle \bar{\boldsymbol{v}}_{\boldsymbol{w}^\perp}, \boldsymbol{u} \rangle$$
$$\ge \kappa \sin(\theta(\boldsymbol{w}, \boldsymbol{v}))$$
$$\ge \frac{\kappa\theta(\boldsymbol{w}, \boldsymbol{v})}{2}$$

where the last inequality holds because $\sin(x) \ge x/2$ for $x \in [0, \pi/2]$. Then, with $\lambda \le \kappa\theta(\boldsymbol{w}, \boldsymbol{v})/4$, Lemma D.4 indicates

$$\langle \boldsymbol{w}', \boldsymbol{v} \rangle \ge \langle \boldsymbol{w}, \boldsymbol{v} \rangle + \frac{\lambda\kappa\theta(\boldsymbol{w}, \boldsymbol{v})}{16}$$

which implies

$$\cos(\theta(\boldsymbol{w}', \boldsymbol{v})) \ge \cos(\theta(\boldsymbol{w}, \boldsymbol{v})) + \frac{\lambda\kappa\theta(\boldsymbol{w}, \boldsymbol{v})}{16}. \tag{4}$$

Because $\cos(t)$ is a decreasing function in $[0, \pi]$, we have that $\theta(\boldsymbol{w}, \boldsymbol{v}) \ge \theta(\boldsymbol{w}', \boldsymbol{v})$. Now, using the trigonometric identity $\cos(x) - \cos(y) = 2\sin((y+x)/2)\sin((y-x)/2)$ gives

$$\cos(\theta(\boldsymbol{w}', \boldsymbol{v})) - \cos(\theta(\boldsymbol{w}, \boldsymbol{v})) = 2\sin\left(\frac{\theta(\boldsymbol{w}, \boldsymbol{v}) + \theta(\boldsymbol{w}', \boldsymbol{v})}{2}\right) \sin\left(\frac{\theta(\boldsymbol{w}, \boldsymbol{v}) - \theta(\boldsymbol{w}', \boldsymbol{v})}{2}\right)$$
$$\le \frac{\theta^2(\boldsymbol{w}, \boldsymbol{v}) - \theta^2(\boldsymbol{w}', \boldsymbol{v})}{2} \tag{5}$$

where the last inequality holds because $\sin(x) \le x$ for $x \in [0, \pi]$. Combining inequality (4) and inequality (5) gives

$$\theta(\boldsymbol{w}', \boldsymbol{v}) \le \theta(\boldsymbol{w}, \boldsymbol{v})\sqrt{1 - \frac{\lambda\kappa}{8\theta(\boldsymbol{w}, \boldsymbol{v})}}$$
$$\overset{(i)}{\le} \theta(\boldsymbol{w}, \boldsymbol{v})\left(1 - \frac{\lambda\kappa}{16\theta(\boldsymbol{w}, \boldsymbol{v})}\right)$$
$$\le \theta(\boldsymbol{w}, \boldsymbol{v}) - \frac{\kappa^2\phi}{64}$$

where inequality (i) holds because $\sqrt{1-x} \leq 1 - x/2$ for $x \in [0, 1)$, and the final result is obtained by taking $\lambda = \kappa\phi/4$. $\qquad\square$

**Lemma D.4** (Correlation Improvement (Diakonikolas et al., 2020a)). *For unit vectors $\boldsymbol{v}, \boldsymbol{w} \in \mathbb{R}^d$, let $\boldsymbol{u} \in \mathbb{R}^d$ be such that $\langle \boldsymbol{u}, \boldsymbol{v} \rangle \geq c$, $\langle \boldsymbol{u}, \boldsymbol{w} \rangle = 0$, and $\|\boldsymbol{u}\|_2 \leq 1$, with $c > 0$. Then, for $\boldsymbol{w}' = \boldsymbol{w} + \lambda\boldsymbol{u}$, with $\lambda \leq c/2$, we have that $\langle \bar{\boldsymbol{w}}', \boldsymbol{v} \rangle \geq \langle \boldsymbol{w}, \boldsymbol{v} \rangle + \lambda c/8$.*

**Lemma D.5** (Lemma 3.4). *Fix two vectors $\boldsymbol{x}, \boldsymbol{v} \in \mathbb{R}^d$ such that $\|\boldsymbol{v}\|_2 = 1$ and $\langle \bar{\boldsymbol{x}}, \boldsymbol{v} \rangle \geq 0$. Then, for any unit vector $\boldsymbol{w} \in \mathbb{R}^d$ that satisfies $\langle \boldsymbol{w}, \bar{\boldsymbol{x}} \rangle < 0$ and $\langle \boldsymbol{w}, \boldsymbol{v} \rangle \geq 0$, it holds that $\theta(\bar{\boldsymbol{w}}_{\boldsymbol{x}^\perp}, \boldsymbol{v}) \leq \theta(\boldsymbol{w}, \boldsymbol{v})$.*

*Proof.* First and foremost, since $\bar{\boldsymbol{w}}_{\boldsymbol{x}^\perp}, \boldsymbol{w}, \boldsymbol{v}$ are all unit vectors, it suffices to show that $\langle \bar{\boldsymbol{w}}_{\boldsymbol{x}^\perp}, \boldsymbol{v} \rangle \geq \langle \boldsymbol{w}, \boldsymbol{v} \rangle$. Observe that we can decompose any vector $\boldsymbol{u} \in \mathbb{R}^d$ into $\boldsymbol{u}_{\boldsymbol{x}}$ on the direction of $\boldsymbol{x}$ and $\boldsymbol{u}_{\boldsymbol{x}^\perp}$ on the orthogonal space of $\boldsymbol{x}$ as illustrated in Figure 2. Therefore, we must have

$$\langle \bar{\boldsymbol{w}}_{\boldsymbol{x}^\perp}, \boldsymbol{v} \rangle = \langle \bar{\boldsymbol{w}}_{\boldsymbol{x}^\perp} - \boldsymbol{w}_{\boldsymbol{x}^\perp} - \boldsymbol{w}_{\boldsymbol{x}} + \boldsymbol{w}, \boldsymbol{v} \rangle$$
$$= \langle \bar{\boldsymbol{w}}_{\boldsymbol{x}^\perp} - \boldsymbol{w}_{\boldsymbol{x}^\perp}, \boldsymbol{v}_{\boldsymbol{x}^\perp} \rangle - \langle \boldsymbol{w}_{\boldsymbol{x}}, \boldsymbol{v}_{\boldsymbol{x}} \rangle + \langle \boldsymbol{w}, \boldsymbol{v} \rangle.$$

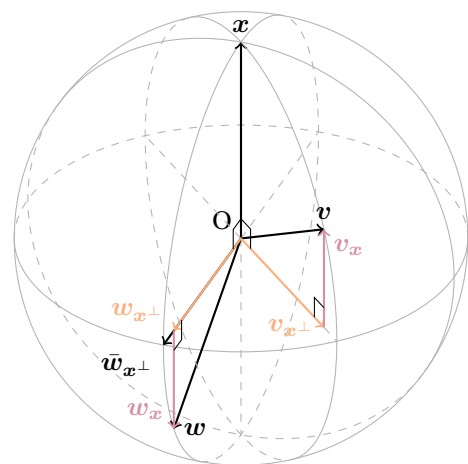

Figure 2: A 3-dimensional visualization of Contractive Projection.

Then, we only need to show $\langle \boldsymbol{w}_{\boldsymbol{x}}, \boldsymbol{v}_{\boldsymbol{x}} \rangle \leq 0$ and $\langle \bar{\boldsymbol{w}}_{\boldsymbol{x}^\perp} - \boldsymbol{w}_{\boldsymbol{x}^\perp}, \boldsymbol{v}_{\boldsymbol{x}^\perp} \rangle \geq 0$. The former inequality holds because we have $\boldsymbol{u}_{\boldsymbol{x}} = \langle \boldsymbol{u}, \bar{\boldsymbol{x}} \rangle \bar{\boldsymbol{x}}$ for any $\boldsymbol{u} \in \mathbb{R}^d$ by definition, while $\langle \boldsymbol{w}, \bar{\boldsymbol{x}} \rangle < 0$ and $\langle \boldsymbol{v}, \bar{\boldsymbol{x}} \rangle \geq 0$ due to our assumption. To prove the latter inequality, note that, because $\langle \boldsymbol{w}_{\boldsymbol{x}}, \boldsymbol{v}_{\boldsymbol{x}} \rangle \leq 0$, it holds that

$$\langle \boldsymbol{w}_{\boldsymbol{x}^\perp}, \boldsymbol{v}_{\boldsymbol{x}^\perp} \rangle \geq \langle \boldsymbol{w}_{\boldsymbol{x}^\perp}, \boldsymbol{v}_{\boldsymbol{x}^\perp} \rangle + \langle \boldsymbol{w}_{\boldsymbol{x}}, \boldsymbol{v}_{\boldsymbol{x}} \rangle$$
$$= \langle \boldsymbol{w}, \boldsymbol{v} \rangle$$
$$\geq 0 \qquad (6)$$

Furthermore, since $\|\boldsymbol{w}_{\boldsymbol{x}^\perp}\|_2 \leq \|\boldsymbol{w}\|_2 = \|\bar{\boldsymbol{w}}_{\boldsymbol{x}^\perp}\|_2$ by the triangle inequality and the unit vector assumption, there must exists an $\alpha \geq 0$ such that $\bar{\boldsymbol{w}}_{\boldsymbol{x}^\perp} - \boldsymbol{w}_{\boldsymbol{x}^\perp} = \alpha\boldsymbol{w}_{\boldsymbol{x}^\perp}$, which, along with inequality (6), implies $\langle \bar{\boldsymbol{w}}_{\boldsymbol{x}^\perp} - \boldsymbol{w}_{\boldsymbol{x}^\perp}, \boldsymbol{v}_{\boldsymbol{x}^\perp} \rangle = \alpha \langle \boldsymbol{w}_{\boldsymbol{x}^\perp}, \boldsymbol{v}_{\boldsymbol{x}^\perp} \rangle \geq 0$. $\qquad\square$

**Lemma D.6** (Wedge Upper Bound). *Let $\mathcal{D}$ be any distribution on $\mathbb{R}^d \times \{0, 1\}$ with $U$-concentrated and anti-concentrated $\mathbf{x}$-marginal, then, for any unit vectors $\boldsymbol{w}, \boldsymbol{v} \in \mathbb{R}^d$, it holds that $\Pr_{(\mathbf{x}, \mathbf{y}) \sim \mathcal{D}}\{\mathbf{x} \in h(\boldsymbol{w}) \backslash h(\boldsymbol{v})\} \leq U\theta(\boldsymbol{w}, \boldsymbol{v})$.*

*Proof.* Let $V$ be the subspace spanned by $\{\boldsymbol{w}, \boldsymbol{v}\}$, where we choose $\boldsymbol{e}_2 = \boldsymbol{w}$ and $\boldsymbol{e}_1 = -\bar{\boldsymbol{v}}_{\boldsymbol{w}^\perp}$ to be a basis when projecting $\mathbf{x} \sim \mathcal{D}$ onto $V$. Suppose $\varphi : \mathbb{R}^d \to \mathbb{R}$ is the density function of $\mathcal{D}_{\mathbf{x}}$ and $\varphi_V$

is its projection on $V$, then we have

$$\Pr_{(\mathbf{x},\mathbf{y})\sim\mathcal{D}}\{\mathbf{x}\in h(\boldsymbol{w})\backslash h(\boldsymbol{v})\} = \int_{\mathbf{x}\in h(\boldsymbol{w})\backslash h(\boldsymbol{v})}\varphi(\mathbf{x})d\mathbf{x}$$

$$= \int_{\mathbf{x}_V\in h(\boldsymbol{w})\backslash h(\boldsymbol{v})}\varphi_V(\mathbf{x}_V)d\mathbf{x}_V$$

$$\overset{(i)}{\le} \int_{\mathbf{x}_V\in h(\boldsymbol{w})\backslash h(\boldsymbol{v})}p(\|\mathbf{x}_V\|_2)d\mathbf{x}_V$$

$$\overset{(ii)}{=} \int_0^{\theta(\boldsymbol{w},\boldsymbol{v})}\int_0^{\infty}\mathrm{r}p(\mathrm{r})d\mathrm{r}d\phi$$

$$\le U\theta(\boldsymbol{w},\boldsymbol{v})$$

where inequality (i) holds because $\mathcal{D}_\mathbf{x}$ is anti-concentrated. Equation (ii) is obtained by transforming the Cartesian coordinates into Polar coordinates with $\mathrm{r} = \|\mathbf{x}_V\|_2$, $\mathrm{x}_1 = \mathrm{r}\cos(\theta(\mathbf{x}_V,\boldsymbol{e}_1))$, $\mathrm{x}_2 = \mathrm{r}\sin(\theta(\mathbf{x}_V,\boldsymbol{e}_1))$, and, hence, $d\mathbf{x}_V = d\mathrm{x}_1 d\mathrm{x}_2 = \mathrm{r}d\mathrm{r}d\phi$. And, the last inequality holds, again, due to the $U$-concentration property. $\qquad\square$

**Proposition D.7** (Proposition 3.5). *Let $\mathcal{D}$ be any distribution on $\mathbb{R}^d \times \{0,1\}$ with centered and $(K, U, L, R)$-well-behaved $\mathbf{x}$-marginal and $\boldsymbol{x} \in \mathbb{R}^d$ be an observation example with non-zero support. If there exists a unit vector $\boldsymbol{v} \in \mathbb{R}^d$ such that $\boldsymbol{x} \in h(\boldsymbol{v})$ and*

$$\Pr_{(\mathbf{x},\mathbf{y})\sim\mathcal{D}}\{\mathbf{y} = 1 \mid \mathbf{x} \in h(\boldsymbol{v})\} \le \epsilon,$$

*then, Algorithm 2 takes*

$$T = 32\pi\epsilon^{-5/4}/\sqrt{2(2K+1)/L},$$

$$\lambda = \sqrt{2(2K+1)/L}\epsilon^{3/4}/4,$$

$$|\hat{\mathcal{D}}| = O(K^2\ln(2T/\delta)/\epsilon),$$

$\boldsymbol{x} \in \mathbb{R}^d$ *as inputs, runs in time at most $\tilde{O}(d\epsilon^{-9/4})$, and outputs a list $\mathcal{W} = \{\boldsymbol{w}^{(0)}, \ldots, \boldsymbol{w}^{(T)}\}$, where there exists a $\boldsymbol{w}^{(t)}$ that satisfies both $\boldsymbol{x} \in h(\boldsymbol{w}^{(t)})$ and*

$$\Pr_{(\mathbf{x},\mathbf{y})\sim\mathcal{D}}\{\mathbf{y} = 1 \mid \mathbf{x} \in h(\boldsymbol{w}^{(t)})\} \le (U\sqrt{2(2K+1)/R^2L} + 1/R)\epsilon^{1/4}$$

*with probability at least $1 - \delta$.*

*Proof.* Obviously, the first condition $\boldsymbol{x} \in h(\boldsymbol{w}^{(t)})$ must hold because the Contractive Projection (line 5-9 of Algorithm 2) guarantees that $\langle\boldsymbol{x}, \boldsymbol{w}^{(i)}\rangle \ge 0$ for each $i \in [T]$.

To prove the second condition, we shall consider three possible cases. If we directly have

$$\Pr\{\mathbf{y} = 1 \mid \mathbf{x} \in h(\boldsymbol{w}^{(i)})\} \le (U\sqrt{2(2K+1)/R^2L} + 1/R)\epsilon^{1/4}$$

in some iteration $i \in [T]$, we are done.

Instead, if some $\boldsymbol{w}^{(i)}$ satisfies $\theta(\boldsymbol{w}^{(i)}, \boldsymbol{v}) \le \sqrt{2(2K+1)/L}\epsilon^{1/4}$, we must have that

$$\Pr\{\mathbf{y} = 1 \mid \mathbf{x} \in h(\boldsymbol{w}^{(i)})\} = \frac{\Pr\{\mathbf{y} = 1 \cap \mathbf{x} \in h(\boldsymbol{w}^{(i)}) \cap \mathbf{x} \in h(\boldsymbol{v})\} + \Pr\{\mathbf{y} = 1 \cap \mathbf{x} \in h(\boldsymbol{w}^{(i)}) \cap \mathbf{x} \notin h(\boldsymbol{v})\}}{\Pr\{\mathbf{x} \in h(\boldsymbol{w}^{(i)})\}}$$

$$\le \frac{\Pr\{\mathbf{y} = 1 \cap \mathbf{x} \in h(\boldsymbol{v})\} + \Pr\{\mathbf{x} \in h(\boldsymbol{w}^{(i)}) \cap \mathbf{x} \notin h(\boldsymbol{v})\}}{\Pr\{\mathbf{x} \in h(\boldsymbol{w}^{(i)})\}}$$

$$\overset{(i)}{\le} \frac{\epsilon + U\sqrt{2(2K+1)/L}\epsilon^{1/4}}{\Pr\{\mathbf{x} \in h(\boldsymbol{w}^{(i)})\}}$$

$$\le (U\sqrt{2(2K+1)/R^2L} + 1/R)\epsilon^{1/4}$$

where inequality (i) results from an application of Lemma D.6 and the fact that $\Pr\{\mathbf{x} \in h(\boldsymbol{v})\} \le 1$, and the last inequality holds because $\epsilon \le \epsilon^{1/4}$ and $\mathcal{D}_\mathbf{x}$ is $R$-Rounded. So $\theta(\boldsymbol{w}^{(i)}, \boldsymbol{v}) \le \sqrt{2(2K+1)/L}\epsilon^{1/4}$ also gives the desired result.

However, we show that the third case, where we have both

$$\Pr\{y = 1 \mid \mathbf{x} \in h(\boldsymbol{w}^{(i)})\} > (U\sqrt{2(2K+1)/R^2 L} + 1/R)\epsilon^{1/4}$$

and $\theta(\boldsymbol{w}^{(i)}, \boldsymbol{v}) > \sqrt{2(2K+1)/L}\epsilon^{1/4}$ in all $T$ iterations, cannot exist by contradiction. Suppose, for the sake of contradiction, both of the inequalities hold for all $i \in [T]$. We argue that the angle between $\boldsymbol{w}^{(i)}$ and $\boldsymbol{v}$ monotonically decreases over iterations by induction, i.e., $\theta(\boldsymbol{w}^{(i)}, \boldsymbol{v}) \leq \theta(\boldsymbol{w}^{(i-1)}, \boldsymbol{v}) - C\epsilon^{5/4}$ for $C = \sqrt{2(2K+1)/L}/64$.

Observe that, for $\boldsymbol{w}^{(0)} = \bar{\boldsymbol{x}}$, the claim is trivially true. Suppose it holds that $\theta(\boldsymbol{w}^{(i)}, \boldsymbol{v}) \leq \theta(\boldsymbol{w}^{(i-1)}, \boldsymbol{v}) - C\epsilon^{5/4}$ for all $\boldsymbol{w}^{(0)}, \dots, \boldsymbol{w}^{(i)}$ and some constant $C > 0$, we wish to show $\theta(\boldsymbol{w}^{(i+1)}, \boldsymbol{v}) \leq \theta(\boldsymbol{w}^{(s)}, \boldsymbol{v}) - C\epsilon^{5/4}$.

Note that $\theta(\boldsymbol{w}^{(0)}, \boldsymbol{v}) \in [0, \pi/2]$ by our assumption and the initialization step, we must have $\theta(\boldsymbol{w}^{(i)}, \boldsymbol{v}) \in [0, \pi/2]$ because of the inductive hypothesis. Then, due to the assumed error lower bound on $\boldsymbol{w}^{(i)}$, we can invoke Lemma D.1 to obtain $\langle \mathbb{E}[-g_{\boldsymbol{w}^{(i)}}(\mathbf{x}, \mathbf{y})], \bar{\boldsymbol{v}}_{\boldsymbol{w}^{(i)}\perp} \rangle \geq \sqrt{\epsilon}$. With $|\hat{\mathcal{D}}| \geq \max(C_0^2 K^2 \ln(T/\delta)/\epsilon, C_0 K \ln(T/\delta)/\sqrt{\epsilon})$, where $C_0 > 0$ is a constant, Lemma G.9 gives

$$\Pr\left\{ \left\langle \underset{\hat{\mathcal{D}}}{\mathbb{E}}[-g_{\boldsymbol{w}^{(i)}}(\mathbf{x}, \mathbf{y})], \bar{\boldsymbol{v}}_{\boldsymbol{w}^{(i)}\perp} \right\rangle < \frac{\sqrt{\epsilon}}{2} \right\} \leq \frac{\delta}{T}. \tag{7}$$

Conditioned on $\langle \mathbb{E}_{\hat{\mathcal{D}}}[-g_{\boldsymbol{w}^{(i)}}(\mathbf{x}, \mathbf{y})], \bar{\boldsymbol{v}}_{\boldsymbol{w}^{(i)}\perp} \rangle \geq \sqrt{\epsilon}/2$, Lemma D.3 indicates that $\theta(\boldsymbol{u}^{(i+1)}, \boldsymbol{v}) \leq \theta(\boldsymbol{w}^{(i)}, \boldsymbol{v}) - C\epsilon^{5/4}$. Notice that, if $\theta(\boldsymbol{u}^{(i+1)}, \boldsymbol{x}) > \pi/2$, Lemma D.5 will guarantee that the contractive projection (line 9) doesn't increase $\theta(\boldsymbol{w}^{(i+1)}, \boldsymbol{v}) \leq \theta(\boldsymbol{u}^{(i+1)}, \boldsymbol{v})$, which completes the inductive proof.

With $T = 32\pi\epsilon^{-5/4}/\sqrt{2(2K+1)/L}$ and $\theta(\boldsymbol{w}^{(0)}, \boldsymbol{v}) \leq \pi/2$, taking a Union Bound on inequality (7) over all $T$ iterations, we must have $\theta(\boldsymbol{w}^{(T)}, \boldsymbol{v}) \leq \sqrt{2(2K+1)/L}\epsilon^{1/4}$ with probability at least $1 - \delta$, which leads to a contradiction.

At last, the sample complexity of Algorithm 2 is obviously $|\hat{\mathcal{D}}| = O(K^2 \ln(2T/\delta)/\epsilon)$ as no new examples are sampled during the run. For time complexity, note that it will take $d|\hat{\mathcal{D}}| = O(K^2 d \ln(T/\delta)/\epsilon)$ time to compute the projected gradient in each iteration, and there are $T = 32\pi\epsilon^{-5/4}/\sqrt{2(2K+1)/L}$ iterations in total. Therefore, the total running time should be $dT|\hat{\mathcal{D}}| = \tilde{O}(d\epsilon^{-9/4})$. $\qquad\square$

With lemma D.7 in hand, we are now ready to prove Theorem 3.1.

**Theorem D.8** (Theorem 3.1). *Let $\mathcal{D}$ be any distribution on $\mathbb{R}^d \times \{0, 1\}$ with centered and $(K, U, L, R)$-well-behaved $\mathbf{x}$-marginal and $\boldsymbol{x} \in \mathbb{R}^d$ be an observation example with non-zero support. If there exists a unit vector $\boldsymbol{v} \in \mathbb{R}^d$ such that $\boldsymbol{x} \in h(\boldsymbol{v})$ and*

$$\Pr_{(\mathbf{x}, \mathbf{y}) \sim \mathcal{D}}\{y = 1 \mid \mathbf{x} \in h(\boldsymbol{v})\} \geq 1 - \epsilon$$

*for some sufficiently small $\epsilon$, then, with at most $\tilde{O}(\epsilon^{-1})$ examples, Algorithm 1 runs in time at most $\tilde{O}(d\epsilon^{-9/4})$ and returns a $\boldsymbol{w}^*$ such that $\boldsymbol{x} \in h(\boldsymbol{w}^*)$ and*

$$\Pr_{(\mathbf{x}, \mathbf{y}) \sim \mathcal{D}}\{y = 1 \mid \mathbf{x} \in h(\boldsymbol{w}^*)\} = 1 - (U\sqrt{2(2K+1)/R^2 L} + 1/R + 1)\epsilon^{1/4}$$

*with probability at least $1 - \delta$.*

*Proof.* First and foremost, let's notice that the labels of the examples in $\hat{\mathcal{D}}_1$ are negated in Algorithm 1. Thus, with $T \geq 32\pi\epsilon^{-5/4}/\sqrt{2(2K+1)/L}$ and $|\hat{\mathcal{D}}_1| \geq CK^2 \ln(2T/\delta)/\epsilon$ for some sufficiently large constant $C$, Proposition D.7 guarantees that there exists a $\boldsymbol{w}' \in \mathcal{W}$ such that

$$\begin{aligned}
\Pr_{(\mathbf{x}, \mathbf{y}) \sim \mathcal{D}}\{y = 1 \mid \mathbf{x} \in h(\boldsymbol{w}')\} &= 1 - \Pr_{(\mathbf{x}, \mathbf{y}) \sim \mathcal{D}}\{y = 0 \mid \mathbf{x} \in h(\boldsymbol{w}')\} \\
&\geq 1 - (U\sqrt{2(2K+1)/R^2 L} + 1/R)\epsilon^{1/4}
\end{aligned} \tag{8}$$

with probability at least $1 - \delta/2$.

Then, applying Corollary G.4 on both $\boldsymbol{w}'$ and $\boldsymbol{w}^*$ with $|\hat{\mathcal{D}}_2| = 32\ln(4T/\delta)/R^2\sqrt{\epsilon}$, we have

$$\Pr_{\hat{\mathcal{D}}_2 \sim \mathcal{D}}\left\{\Pr_{\hat{\mathcal{D}}_2}\{y = 1 \mid \mathbf{x} \in h(\boldsymbol{w}')\} < \Pr_{\mathcal{D}}\{y = 1 \mid \mathbf{x} \in h(\boldsymbol{w}')\} - \frac{\epsilon^{1/4}}{2}\right\} \leq \frac{\delta}{2T}$$

or

$$\Pr_{\hat{\mathcal{D}}_2 \sim \mathcal{D}}\left\{\Pr_{\hat{\mathcal{D}}_2}\{y = 1 \mid \mathbf{x} \in h(\boldsymbol{w}^*)\} > \Pr_{\mathcal{D}}\{y = 1 \mid \mathbf{x} \in h(\boldsymbol{w}^*)\} + \frac{\epsilon^{1/4}}{2}\right\} \leq \frac{\delta}{2T}.$$

We take a Union Bound over $\mathcal{W}$ to make sure the above two inequality holds simultaneously. Also, because of empirical minimization step (Line 7) of Algorithm 1, we must have

$$\Pr_{\hat{\mathcal{D}}_2}\{y = 1 \mid \mathbf{x} \in h(\boldsymbol{w}^*)\} \geq \Pr_{\hat{\mathcal{D}}_2}\{y = 1 \mid \mathbf{x} \in h(\boldsymbol{w}')\},$$

which further implies that

$$\Pr\left\{\Pr_{\mathcal{D}}\{y = 1 \mid \mathbf{x} \in h(\boldsymbol{w}^*)\} \geq \Pr_{\mathcal{D}}\{y = 1 \mid \mathbf{x} \in h(\boldsymbol{w}')\} - \epsilon^{1/4}\right\} \leq \frac{\delta}{2}. \qquad (9)$$

Finally, take another Union Bound over inequalities (8) and (9), we can conclude that

$$\Pr_{(\mathbf{x},y)\sim\mathcal{D}}\{y = 1 \mid \mathbf{x} \in h(\boldsymbol{w}')\} \geq 1 - (U\sqrt{2(2K+1)/R^2L} + 1/R + 1)\epsilon^{1/4}$$

with probability at least $1 - \delta$.

Obviously, the sample complexity is $O(\hat{\mathcal{D}}_1 + \hat{\mathcal{D}}_2) = \tilde{O}(\epsilon^{-1})$. For the time complexity, note first that step 4 of Algorithm 2 takes $O(d|\hat{\mathcal{D}}_1|) = \tilde{O}(\epsilon^{-1})$ time to run. Hence, the running time of Algorithm 2 is then $\tilde{O}(d\epsilon^{-9/4})$ as $T = O(\epsilon^{-5/4})$. Similarly, estimating the conditional probability for each $\boldsymbol{w} \in \mathcal{W}$ at step 7 in Algorithm 1 takes $O(d|\hat{\mathcal{D}}_2|) = \tilde{O}(\epsilon^{-1/2})$ time to run. Thus, it takes $\tilde{O}(d\epsilon^{-7/4})$ time to finish step 7. Overall, the running time of Algorithm will be $\tilde{O}(d\epsilon^{-9/4})$. $\qquad\square$

# E   ANALYSIS OF ALGORITHM 3

**Theorem E.1** (Theorem 4.2). *Let $\mathcal{D}$ be a distribution on $\mathbb{R}^d \times \{0,1\}$ with $(K, U, L, R)$-well-behaved $\mathbf{x}$-marginal, $\mathcal{C}$ be a class of sparse linear classifiers on $\mathbb{R}^d \times \{0,1\}$ with sparsity $s = O(1)$, and $\boldsymbol{x} \in \mathbb{R}^d$ be a query point. If there exists a unit vector $\boldsymbol{v} \in \mathbb{R}^d$ and a $c \in \mathcal{C}$ such that $\boldsymbol{x} \in h(\boldsymbol{v})$ and*

$$\Pr_{(\mathbf{x},y)\sim\mathcal{D}}\{c(\mathbf{x}) \neq y \mid \mathbf{x} \in h(\boldsymbol{v})\} \leq \mathrm{opt}$$

*for some sufficiently small* opt*, then, with at most* $\mathrm{poly}(d, 1/\epsilon, 1/\delta)$ *examples, Algorithm 3 will run in time* $\mathrm{poly}(d, 1/\epsilon, 1/\delta)$ *and find a classifier $c^*$ and a halfspace $h(\boldsymbol{w}^*)$ such that $\boldsymbol{x} \in h(\boldsymbol{w}^*)$ and*

$$\Pr_{(\mathbf{x},y)\sim\mathcal{D}}\{c^*(\mathbf{x}) \neq y \mid \mathbf{x} \in h(\boldsymbol{w}^*)\} = O(\mathrm{opt}^{1/4} + \epsilon)$$

*with probability at least $1 - \delta$.*

*Proof.* We first show that the returned list of Algorithm 4 will contain a classifier $c' \in L$ such that $\min_{\boldsymbol{w}} \Pr_{(\mathbf{x},y)\sim\mathcal{D}}\{c'(\mathbf{x}) \neq y \mid \mathbf{x} \in h(\boldsymbol{w})\} \leq \mathrm{opt} + \epsilon^4$.

We decompose the distribution $\mathcal{D}$ into a convex combination of an inlier distribution $\mathcal{D}^*$ and a outlier distribution $\tilde{\mathcal{D}}$ in the following way. Let $\mathcal{D}^*$ be a distribution on $\mathbb{R}^d \times \{0,1\}$ with well-behaved $\mathbf{x}$-marginal such that its labels are generated by $c(\mathbf{x})$, while $\tilde{\mathcal{D}}$ will be a distribution on $\mathbb{R}^d \times \{0,1\}$ with the same $\mathbf{x}$-marginals to be specified later. Observe that, since $\Pr\{c(\mathbf{x}) \neq y \mid \mathbf{x} \in h(\boldsymbol{v})\} \leq \mathrm{opt}$ and $\Pr\{\mathbf{x} \in h(\boldsymbol{v})\} \geq R$ by Definition C.1, at least $R(1 - \mathrm{opt})$ fraction (weighted by the density of $\mathcal{D}_{\mathbf{x}}$) of the labels of $\mathcal{D}$ are consistent with $c(\mathbf{x})$. Therefore, there must exist some $\alpha \geq R(1 - \mathrm{opt})$ such that the labels of $\mathcal{D}_{\mathbf{x}}$ can be generated by selecting labels from $\mathcal{D}^*$ with probability mass $\alpha$ and from $\tilde{\mathcal{D}}$, given by $\mathcal{D}$ conditioned on falling outside the support of $\mathcal{D}^*$, with probability mass $1 - \alpha$, namely $\mathcal{D} = \alpha\mathcal{D}^* + (1 - \alpha)\tilde{\mathcal{D}}$.

Hence, with $m = O((s\log d + \log\frac{2}{\delta})/\epsilon^4)$ examples, we can inovke Theorem B.2 (and Definition B.1) to conclude that there exists a classifier $c' \in L$ such that $\min_{\boldsymbol{w}} \Pr\{c'(\mathbf{x}) \neq y \mid \mathbf{x} \in h(\boldsymbol{w})\} \leq \mathrm{opt} + \epsilon^4$

with probability at least $1-\delta/2$. Meanwhile, it is easy to see that Algorithm 4 runs in $\mathrm{poly}(d, 1/\epsilon, 1/\delta)$ time since $\alpha$ is a constant.

Then, by Theorem 3.1 and the parameters described at Line 8 of Algorithm 3, we have that

$$\Pr_{(\mathbf{x},\mathrm{y})\sim\mathcal{D}}\{c'(\mathbf{x}) = \mathrm{y} \mid \mathbf{x} \in h(\boldsymbol{w}^{(c')})\} = O(\mathrm{opt}^{1/4} + \epsilon)$$

with probability at least $1 - \delta/2\,|L|$. Applying Corollary G.4 (conditional Chernoff Bound) as well as a Union Bound over all candidates in $\mathcal{W}$ (as defined in Algorithm 3) to the empirical estimation (Line 11) with $|\hat{\mathcal{D}}| = 8\ln\left(8\,|L|\,/\delta\right)/R^2\epsilon^2$ and $|L| = O((md)^s)$ gives

$$\Pr_{(\mathbf{x},\mathrm{y})\sim\mathcal{D}}\{c^*(\mathbf{x}) = \mathrm{y} \mid \mathbf{x} \in h(\boldsymbol{w}^*)\} = O(\mathrm{opt}^{1/4} + \epsilon)$$

with probability at least $1 - \delta/2$. Finally, taking another Union Bound over the call of Algorithm 4 and the rest of the algorithm gives the desired result. □

# F   DETAILS OF EXPERIMENTS

For convenience, we also list our experiment results here.

Table 3: Test error rates. TOTAL and LIST denote the number of examples used in the entire training process (Algorithm 3 and baseline models) and the list learning (Algorithm 4) only. The models from left (LOGREG) to right (PERS) are logistic regression, SVM with Linear, RBF kernel, XGBoost tree, random forest, ERM sparse classifier (SPARSE), and personalized prediction (PERS) respectively. * indicates statistically significant improvement with 95% confidence (over SPARSE for PERS, and over PERS for the other baselines). For Pima and Hepa, PERS obtains improvement over SPARSE with 85% confidence, and the difference from the other baselines is not significant at this level.

| D/S | TOTAL | LIST | DIM | LOGREG | LIN | RBF | XGB | FOREST | SPARSE | PERS |
|---|---|---|---|---|---|---|---|---|---|---|
| HABE | 204 | 204 | 3 | .2647 | .2647 | .2941 | .3529 | .3039 | .2745 | .2745 |
| PIMA | 512 | 192 | 8 | .2461 | .25 | .2344 | .2344 | .2304 | .2852 | .2461 |
| HEPA | 103 | 103 | 20 | .1538 | .1538 | .1346 | .2115 | .1538 | .2308 | .1538 |
| HYPO | 2109 | 64 | 20 | .0199* | .019* | .0285 | .0133* | .0142* | .0579 | .0379* |
| WDBC | 379 | 48 | 30 | .0368 | .0474 | .0421 | .0421 | .0579 | .0789 | .0474* |

Overall, we simply evaluated seven algorithms, i.e. logistic regression, SVM with linear kernel, SVM with RBF kernel, XGBoost, random forest, ERM sparse classifier (Algorithm 4 with an additional evaluation, explained later), and personalized prediction (Algorithm 3), over five tabular datasets of binary classification task. Since these datasets are all binary labeled, we only focused on the 0-1 classification loss as our experiment metric.

The first five baselines represent a variety of methods for real-world (moderate-to-small data) classification tasks, exemplified by the UCI benchmarks, while the ERM sparse classifier is used as a special baseline to demonstrate that the sparse classifier selected with the help of our reference class algorithm (Algorithm 1) could actually outperform the ERM of the sparse classifiers returned by the robust classification algorithm (Algorithm 4). In particular, we simply run the robust list learner and select the classifier in its returned list obtaining the highest accuracy using the same training dataset (i.e., an *Empirical Risk Minimizer (ERM)*). This special baseline aims to verify that our approach indeed improves the performance of stand alone sparse linear classifiers by learning a corresponding homogeneous halfspace subset for each of them.

For data cleaning, we used one-hot encodings for binary categorical features. Then, we centered and normalized the features so that every feature has mean zero and variance one. For each dataset, we randomly selected $2/3$ of the data as a training sample and use the remaining data as our test set. For all datasets, we use 2-sparse linear classifiers for our personalized prediction scheme.

Since LOGREG, LIN, RBF, XGB, and FOREST are deterministic algorithms, we only ran one trial of each. However, due to the excessively high computational cost of list learning and our limited computation resources (4×NVIDIA A40), we have to randomly sample a small subset from the

training dataset for Algorithm 4, similar to Hainline et al. (2019). We do this because, for example, running the list learning algorithm with sparsity two on a 128-sample of dimension 30 is already prohibitively expensive, i.e., takes $\approx 2300$ hours on Wdbc dataset.

Since the subsets are too small for the theoretical guarantees of probabilistic stability to hold, a good (sparse) classifier may not be included in the list in some trials, and the accuracy may have high variance. Thus, we ran 50 trials for each of these two algorithms, and reported the median of the 50 observed losses in Table 3. This explains why our method becomes less competitive as the data dimension increases due to our sub-sampling strategy. Specifically, for Haberman, our classifier is not actually "sparse" as the sparsity almost equals the data dimension. More importantly, we can afford to run the robust list learning algorithm on the whole training dataset because of the low dimension. Indeed, our approach performs the best for this dataset as shown in Table 3. Because of these limitations, our experiment results may not be able to exhibit the full potential of the personalized prediction scheme.

## G  CONCENTRATION TOOLS

**Fact G.1** (Gaussian properties). *Let* $z \sim \mathcal{N}(0, \sigma^2)$, *we have* $\|z\|_{\psi_2} = \sqrt{8/3}\sigma$ *and* $\Pr\{z \geq t\} \leq e^{-t^2/2\sigma^2}$.

**Definition G.2** (Sub-exponential norm Vershynin (2018)). *For any random variable* $x \sim \mathcal{D}$ *on* $\mathbb{R}$, *we define* $\|x\|_{\psi_1} = \inf\left\{t > 0 \mid \mathbb{E}_{x \sim \mathcal{D}}[e^{|x|/t}] \leq 2\right\}$.

**Lemma G.3** (Chernoff Bound of Additive Form). *Let* $x_1, \ldots, x_m$ *be a sequence of* $m$ *independent Bernoulli trials, each with probability of success* $\mathbb{E}[x_i] = p$, *then with* $t \in [0, 1]$, *there is*

$$\Pr\left\{\left|\frac{1}{m}\sum_{i=1}^{m} x_i - p\right| > t\right\} \leq 2e^{-2mt^2}.$$

**Corollary G.4** (Conditional Chernoff Bound of Additive Form). *Let* $\mathcal{D}$ *be any distribution on* $\mathbb{R}^d \times \{0, 1\}$ *with centered sub-exponential* $x$-*marginals, and* $S$ *be any event such that* $\Pr_{\mathcal{D}}\{x \in S\} \geq R$ *for some constant* $R \in (0, 1]$. *Given* $\hat{\mathcal{D}} = \left\{(y^{(1)}, x^{(1)}), \ldots, (y^{(m)}, x^{(m)})\right\}$ *sampled i.i.d. from* $\mathcal{D}$, *for every* $t \in [0, 1]$, *we have*

$$\Pr_{\hat{\mathcal{D}} \sim \mathcal{D}}\left\{\left|\Pr_{\hat{\mathcal{D}}}\{y = 1 \mid x \in S\} - \Pr_{\mathcal{D}}\{y = 1 \mid x \in S\}\right| > t\right\} \leq 4e^{-mt^2R^2/8}$$

*Proof.* Observe that, by lemma G.3, we have

$$\Pr_{\hat{\mathcal{D}} \sim \mathcal{D}}\left\{\left|\Pr_{\hat{\mathcal{D}}}\{y = 1 \cap x \in S\} - \Pr_{\mathcal{D}}\{y = 1 \cap x \in S\}\right| > t_1\right\} \leq 2e^{-2mt_1^2}$$

as well as

$$\Pr_{\hat{\mathcal{D}} \sim \mathcal{D}}\left\{\left|\Pr_{\hat{\mathcal{D}}}\{x \in S\} - \Pr_{\mathcal{D}}\{x \in S\}\right| > t_1\right\} \leq 2e^{-2mt_1^2}$$

for some $t_1 \geq 0$. Suppose $R \geq 2t_1$. Taking a union bound over the above inequalities gives

$$1 - 4e^{-2mt_1^2} \leq \Pr_{\hat{\mathcal{D}} \sim \mathcal{D}}\left\{\frac{\Pr_{\mathcal{D}}\{y = 1 \cap x \in S\} - t_1}{\Pr_{\mathcal{D}}\{x \in S\} + t_1} \leq \frac{\Pr_{\hat{\mathcal{D}}}\{y = 1 \cap x \in S\}}{\Pr_{\hat{\mathcal{D}}}\{x \in S\}} \leq \frac{\Pr_{\mathcal{D}}\{y = 1 \cap x \in S\} + t_1}{\Pr_{\mathcal{D}}\{x \in S\} - t_1}\right\}$$

$$\overset{(i)}{\leq} \Pr_{\hat{\mathcal{D}} \sim \mathcal{D}}\left\{\frac{\Pr_{\mathcal{D}}\{y = 1 \cap x \in S\} - 2t_1}{\Pr_{\mathcal{D}}\{x \in S\}} \leq \frac{\Pr_{\hat{\mathcal{D}}}\{y = 1 \cap x \in S\}}{\Pr_{\hat{\mathcal{D}}}\{x \in S\}} \leq \frac{\Pr_{\mathcal{D}}\{y = 1 \cap x \in S\} + 4t_1}{\Pr_{\mathcal{D}}\{x \in S\}}\right\}$$

$$\leq \Pr_{\hat{\mathcal{D}} \sim \mathcal{D}}\left\{\frac{\Pr_{\mathcal{D}}\{y = 1 \cap x \in S\} - 4t_1}{\Pr_{\mathcal{D}}\{x \in S\}} \leq \frac{\Pr_{\hat{\mathcal{D}}}\{y = 1 \cap x \in S\}}{\Pr_{\hat{\mathcal{D}}}\{x \in S\}} \leq \frac{\Pr_{\mathcal{D}}\{y = 1 \cap x \in S\} + 4t_1}{\Pr_{\mathcal{D}}\{x \in S\}}\right\}$$

$$= \Pr_{\hat{\mathcal{D}} \sim \mathcal{D}}\left\{\left|\Pr_{\hat{\mathcal{D}}}\{y = 1 \mid x \in S\} - \Pr_{\mathcal{D}}\{y = 1 \mid x \in S\}\right| \leq \frac{4t_1}{\Pr_{\mathcal{D}}\{x \in S\}}\right\}$$

$$\leq \Pr_{\hat{\mathcal{D}} \sim \mathcal{D}}\left\{\left|\Pr_{\hat{\mathcal{D}}}\{y = 1 \mid x \in S\} - \Pr_{\mathcal{D}}\{y = 1 \mid x \in S\}\right| \leq \frac{4t_1}{R}\right\}$$

where inequality (i) holds because, when $a = \Pr\{y = 1 \cap x \in S\} - t_1$ and $b = \Pr\{x \in S\} + t_1$, we can apply the inequality $\frac{a}{b} \leq \frac{a+t_1}{b+t_1}$ to the first term, and, when $a = \Pr\{y = 1 \cap x \in S\}$ and $b = \Pr\{x \in S\} \geq R \geq 2t_1$, we can apply the inequality $\frac{a+t_1}{b-t_1} \leq \frac{a+4t_1}{b}$ to the third term. The final inequality holds because of our assumption that $\Pr\{x \in S\} \geq R$. Finally, taking $t = 4t_1/R$ gives the desired result. $\square$

**Lemma G.5** (Bernstein's Inequality). *Let* $x_1, \ldots, x_m$ *be a sequence of* $m$ *independent, mean zero, sub-exponential random variables. Then, for some absolute constant* $C > 0$ *and every* $t \geq 0$*, we have*

$$\Pr\left\{ \frac{1}{m} \sum_{i=1}^{m} x_i \geq t \right\} \leq \exp\left( -C \min\left( \frac{t^2}{K^2}, \frac{t}{K} \right) m \right)$$

*where* $K = \max_i \|x_i\|_{\psi_1}$.

**Lemma G.6** (Proposition 2.7.1 in Vershynin (2018)). *Let* $\mathcal{D}$ *be any distribution on* $\mathbb{R}$ *such that* $\hat{\|}x\hat{\|}_p \leq Kp$ *for some constant* $K \geq 0$*, then there exists some absolute constant* $C$ *such that* $\|x\|_{\psi_1} \leq CK$.

**Lemma G.7.** *Let* $\mathcal{D}$ *be any distribution on* $\mathbb{R}^d \times \{0, 1\}$ *with* **x***-marginal such that* $\|\langle \mathbf{x}, \boldsymbol{u} \rangle\|_{\psi_1} \leq K$ *for some unit vector* $\boldsymbol{u} \in \mathbb{R}^d$*. For any event* $T \subseteq \mathbb{R}^d$*, we have* $\|y \cdot \langle \mathbf{x}, \boldsymbol{u} \rangle \mathbb{1}\{\mathbf{x} \in T\}\|_{\psi_1} \leq K$.

*Proof.* Because $y$ and $\mathbb{1}\{\mathbf{x} \in T\}$ are boolean valued, we have

$$\mathbb{E}[\exp\left(|y \cdot \langle \mathbf{x}, \boldsymbol{u} \rangle \mathbb{1}\{\mathbf{x} \in T\}| / K\right)] \leq \mathbb{E}[\exp\left(|\langle \mathbf{x}, \boldsymbol{u} \rangle| / K\right)]$$
$$\overset{(i)}{\leq} \mathbb{E}[\exp\left(|\langle \mathbf{x}, \boldsymbol{u} \rangle| / \|\langle \mathbf{x}, \boldsymbol{u} \rangle\|_{\psi_1}\right)]$$
$$\leq 2$$

where inequality (i) holds because $\mathbb{E}[\exp\left(|\langle \mathbf{x}, \boldsymbol{u} \rangle| / t\right)]$ is a decreasing function of $t$, and the last inequality is by Definition G.2. Also, by the same definition, the above inequality implies the claimed result. $\qquad\square$

**Lemma G.8** (Exercise 2.7.10 in Vershynin (2018)). *If* $x \sim \mathcal{D}$ *is a sub-exponential random variable on* $\mathbb{R}$ *such that* $\|x\|_{\psi_1} \leq K$*, then there exists some absolute constant* $C$ *such that* $\|x - \mathbb{E}_{\mathcal{D}}[x]\|_{\psi_1} \leq CK$.

**Corollary G.9.** *Let* $\mathcal{D}$ *be any distribution on* $\mathbb{R}^d \times \{0, 1\}$ *with* $K$*-bounded* **x***-marginal and* $\hat{\mathcal{D}} \overset{i.i.d.}{\sim} \mathcal{D}$ *be an* $m$*-sample. Define* $g_{\boldsymbol{w}}(\mathbf{x}, y) = y \cdot \mathbf{x}_{\boldsymbol{w}^\perp} \mathbb{1}\{\mathbf{x} \in h(\boldsymbol{w})\}$*. For any fixed* $\boldsymbol{v}, \boldsymbol{w} \in \mathbb{R}^d$*, it holds that*

$$\Pr\left\{ \left| \left\langle \underset{\hat{\mathcal{D}}}{\mathbb{E}}[g_{\boldsymbol{w}}(\mathbf{x}, y)] - \underset{\mathcal{D}}{\mathbb{E}}[g_{\boldsymbol{w}}(\mathbf{x}, y)], \bar{\boldsymbol{v}}_{\boldsymbol{w}^\perp} \right\rangle \right| > t \right\} \leq \exp\left( -\min\left( \frac{t^2}{C^2 K^2}, \frac{t}{CK} \right) m \right)$$

*where* $C > 0$ *is an absolute constant.*

*Proof.* Let 's first notice that $\langle \mathbf{x}_{\boldsymbol{w}^\perp}, \bar{\boldsymbol{v}}_{\boldsymbol{w}^\perp} \rangle = \langle \mathbf{x}, \bar{\boldsymbol{v}}_{\boldsymbol{w}^\perp} \rangle$ due to the definition of projection. Then, by Lemma G.6 and our distributional assumption, we have $\|\langle \mathbf{x}_{\boldsymbol{w}^\perp}, \bar{\boldsymbol{v}}_{\boldsymbol{w}^\perp} \rangle\|_{\psi_1} \leq C_0 K$ for some constant $C_0 > 0$. Now, according to Lemma G.7 and G.8, it holds that $\|\langle g_{\boldsymbol{w}}(\mathbf{x}, y), \bar{\boldsymbol{v}}_{\boldsymbol{w}^\perp} \rangle - \mathbb{E}[\langle g_{\boldsymbol{w}}(\mathbf{x}, y), \bar{\boldsymbol{v}}_{\boldsymbol{w}^\perp} \rangle]\|_{\psi_1} \leq CK$ for some constant $C \geq 0$. At last, applying Lemma G.5 on $\langle g_{\boldsymbol{w}}(\mathbf{x}, y) - \mathbb{E}[g_{\boldsymbol{w}}(\mathbf{x}, y)], \bar{\boldsymbol{v}}_{\boldsymbol{w}^\perp} \rangle$ gives the claimed tail bound. $\qquad\square$

