# OpenReview forum: "Personalized Prediction By Learning Halfspace Reference Classes Under Well-Behaved Distribution"
_ICLR.cc/2026/Conference — Submitted to ICLR 2026_

### Official Review · Reviewer_8waY · 2025-10-31

**Soundness:** 3
**Presentation:** 3
**Contribution:** 2
**Rating:** 2
**Confidence:** 3

**Summary:**

The paper studies a new personalized prediction problem where the objective is defined by some PAC-like inequality. Specifically, the hypothesis class is considered to be the class of sparse linear functions and the "personalization" constraint is modeled by halfspaces. The paper draws a connection with the reference class learning problem and builds algorithm for the personalized prediction problem based on some improvement of algorithms therein.

**Strengths:**

The paper is very technical but well-written. In my opinion, the contribution of the paper lies in two-fold:

- The formulation of the personalized prediction problem. The paper proposes a new formulation, and it has not been studied in the literature.
- The formulation draws connection with the reference class learning problem. The paper twists Huang and Juba (2025) and improves the algorithm and adapts it to the context of personalized prediction problem.

**Weaknesses:**

For the context of writing this review, there is a clear stream of papers working on the topic of this paper: Diakonikolas et al., 2020b;c; 2021; 2022; 2024; Juba and Li (2020); Huang and Juba (2025). I, as the reviewer, am not familiar with this stream of work; I have moderate exposure to learning theory and write papers using learning theory. So my judgment could be, on the one hand, biased by the limited scope of my own background, but on the other hand, it could be representative of the general ML people who do research at the intersection of theory and methodology.

In this light, my main concerns are:
- The studied formulation is too niche and doesn't provide much practical insight. Personalized prediction is nevertheless an important problem, but the algorithms and analyses in this paper don't provide much insight for practical usage of these algorithms.
- The theoretical contribution along the existing literature above is unclear from my reading of the paper. I understand that the problem studied is new and can't be resolved by directly applying the existing algorithms. But the question is whether the new treatment and analysis in this paper is generalizable to other theory problems, say, to be used for other problems to make it possible/improve the analysis therein.

To this end, I feel the paper is more suitable for COLT, where the technical contribution can be better accessed.

**Questions:**

See above.

---

> ### Author Response · Authors · 2025-11-18
>
> We thank the reviewer for the valuable advice! We respond to the concerns of the reviewer as follows:
>
> * Could you be more specific on which parts of our problem formulation are niche? More importantly, could you provide concrete reasons why you think our formulation is niche or does not provide enough practical insights?
>
> * First of all, we have highlighted our contributions in Section 1.2 from both theoretical and practical aspects. As you mentioned, the problem we studied is new and can't be resolved by directly applying the existing algorithms. Therefore, the major contribution is the first PAC-learning algorithm for this new problem. Could you be more specific on which parts of the contributions are not clear in your opinion? For the other part of the concern, the answer usually depends on what specific theory problem you are considering and is not necessarily possible in general. Could you be more specific on what theory problems you were referring to?

---

### Official Review · Reviewer_cGFC · 2025-11-03

**Soundness:** 4
**Presentation:** 4
**Contribution:** 3
**Rating:** 6
**Confidence:** 3

**Summary:**

This paper proposes an algorithm for personalized prediction for more interpretable learning by focusing on a subset of feature points. Its algorithm combines a robust list learner for the class of sparse linear classifiers and a reference class learner. Specifically, the learning is based on projected gradient descent with an analysis that each step of the update will improve the correlation between the candidate classifier and the assumed ground truth that has small prediction error with respect to the underlying distribution. It is shown that the algorithm runs in polynomial time with respect to parameters $d, 1/\epsilon, 1/\delta$.

**Strengths:**

The paper uses (mostly) existing techniques and builds an algorithm to solve the personalized prediction problem. The algorithm works under the more general distributional assumptions, i.e. well-behaved marginal distribution that resembles uniform, Gaussian, and many log-concave distributions, by scarifying a small amount of accuracy. From the technical perspective, it improves upon the existing algorithmic guarantees over that of Huang and Juba (2025) by showing the existence of the candidate point $x$ in predicted set $S$.

From a broader perspective, the paper builds a learner for a more interpretable model, using sparse linear classifiers. It maintains the accuracy at the same time by personalizing on a specific query $x$. The theory is sound and the algorithms runs in polynomial time with empirical study on benchmark data sets.

**Weaknesses:**

Comparing to its prior work, e.g. Huang and Juba (2025), Diakonikolas et al. (2022), the contribution seems incremental. Besides a slight different setting, i.e. query $x' \in S$ and well-behaved distribution, the algorithm and analysis resume those were already in prior works. Hence, it is more like a new application of the existing techniques in a personalized learning setting. I wonder how broad the applications of the proposed algorithm can be.

**Questions:**

See weaknesses. What is the essential contribution for the proposed algorithms in personalized learning setting, besides interpretability? What is the key difference between personalized learners and simple invoking the traditional PAC learners for sparse linear classifiers? Some examples on specific $x$ where traditional PAC learner would fail yet the personalized learner can accurately and efficiently predict it, would help much more.

---

> ### Author Response · Authors · 2025-11-18
>
> We thank the reviewer for the valuable advice! We will address the weaknesses and answer the questions in the following:
>
> # Weaknesses
> Note that our gradient descent algorithm, the main component of the personalized prediction, is different from the counterparts in both [1] and [7] because of the contractive projection. Although we did utilize some of the results in [1] and [7], the majority of our analysis is new and not similar to that of [1, 7] due to the algorithmic change. Please see Section 1.3 for more detailed algorithmic difference between our work and theirs. Given these differences, I would argue that our contribution is not merely incremental.
>
> # Questions
>
> As you mentioned, we are considering a different problem setting from traditional learning scheme. Briefly speaking, we are trying to find a classifier for each incoming query, while the the traditional learning scheme seeks to learn a universal classifier for all future queries. Therefore, besides the interpretability, we proposed the first PAC learning algorithm for a personalized classification learning model that is different from traditional classification. Please see Section 1.1 \& 1.2 for more discussion of how our problem setting is different and what our contributions are. If you have any specific questions regarding our presentation in those sections, please let us know, we are happy to revise our paper based on your suggestions.
>
> __Please see the common answer for the references__

---

### Official Review · Reviewer_Hoge · 2025-11-04

**Soundness:** 3
**Presentation:** 3
**Contribution:** 2
**Rating:** 4
**Confidence:** 4

**Summary:**

this paper propose a personalized prediction framework for sparse binary classification, where instead of learning one universal model, the approach learns a dedicated classifier for each query by focusing on a relevant subpopulation that best represents the query. The classifier is also designed to be sparse and interpretable. The key idea is that focusing on such subpopulations, characterized by homogeneous halfspaces, makes learning and interpretation easier, since simpler representations can capture local structures within a subset of the data more effectively than a single global model. The authors also provide algorithms with theoretical guarantee and conduct numerical experiments.

**Strengths:**

1. The paper is well written and clearly structured, making both the motivation and technical developments easy to follow. The exposition of definitions, algorithms, and proofs is systematic and readable.

2. The paper provides algorithms generating useful results with theoretical guarantees, which is a non-trivial extension of existing conditional learning results to the personalized prediction setting.

**Weaknesses:**

The numerical section is a major weakness of the paper. Even after carefully reading the appendix, it remains difficult to understand what the experiments are actually demonstrating. This lack of clarity significantly undermines the overall quality of the paper, especially given that the theoretical section is otherwise well written.

1. From a conceptual standpoint, it is unclear how the numerical experiments connect to the theoretical results. Simply stating that “the results are lower” is too vague to be meaningful. The authors should explain why a lower value supports the theorem, and how the intuition from the theoretical analysis is reflected in the empirical behavior.

2. The authors also fail to properly explain Table 2, which appears to be the central empirical result. The statement “* indicates statistically significant improvement with 95% confidence (over SPARSE for PERS, and over PERS for the other baselines)” is confusing because the metric and experimental setup are never formally defined. Moreover, in the HYPO dataset, XGB (0.142*) and PERS (0.379*) differ by a large margin (nearly a factor of two) yet both are marked as significant. Without a clear description of the evaluation protocol, metrics, and comparison criteria, the empirical results are difficult to interpret and fail to substantiate the theoretical claims.

I recommend that the authors improve the numerical section in the appendix.

**Questions:**

The authors should also provide intuition on why the chosen UCI datasets are appropriate for supporting the theoretical results (provide reference if already validated in other works). Since the theory relies on assumptions such as well-behaved distributions and sparse feature relevance, it would be helpful to include illustrations or diagnostic plots showing whether the datasets approximately satisfy these assumptions (e.g., marginal distributions, sparsity patterns, or subspace structure).

Additionally, what is the experimental setup? Readers need information about the parameter settings, number of trials, and criteria for statistical significance to properly evaluate the experiments.

---

> ### Author Response · Authors · 2025-11-18
>
> We thank the reviewer for the valuable advice! We will address the weaknesses and answer the questions in the following:
>
> # Weaknesses
>
> We apologize for the confusion regarding the numerical section. Overall, we simply evaluated seven algorithms, i.e. logistic regression, SVM with linear kernel, SVM with RBF kernel, XGBoost, random forest, ERM sparse classifier (Algorithm 4 in Appendix B with an additional evaluation, explained at the end of Appendix F), and personalized prediction (Algorithm 3 in Section 4.1), over five tabular datasets of binary classification task. Since these datasets are all binary labeled, we only focused on the 0-1 classification loss as our experiment metric. The first five algorithms were used as general baselines to compare with personalized prediction, while the ERM sparse classifier is used as a special baseline to demonstrate that the sparse classifier selected with the help of our reference class algorithm (Algorithm 1 in Section 3.1) could actually outperform the ERM of the sparse classifiers returned by the robust classification algorithm (Algorithm 4 in Appendix B).
>
> We address the specific concerns raised by the reviewer below:
>
> * The comparison to ERM serves a distinct purpose from the comparison to the other baselines: think of it as an ablation, demonstrating that the personalization is helpful compared to simply using a sparse linear classifier. Hence why the statistical significance mark is used here to indicate whether PERS has significantly lower error than ERM.
>
> * The other baselines represent a variety of methods for real-world (moderate-to-small data) classification tasks, exemplified by the UCI benchmarks. Although the theorems give some general conditions under which the algorithm may be expected to perform well, this is an orthogonal question to whether the proposed method actually attains accuracy that is competitive with other standard classification methods for these standard benchmark tasks. The theorems are worst-case guarantees, subject to the distributional niceness assumptions, and both our methods and the baselines may often outperform such guarantees. Indeed, for example, these domains may contain structure that the various methods (both ours and the baseline) might exploit, that is not captured by the theorems, and the question is whether that generally happens. At the same time, the upper bounds in the theorems can hardly be verified through empirical experiments due to the computational challenge in testing the distributional assumptions (we will discuss more in response to the Questions part). That is, in summary, the experiments are complementary to the theorems, indicating how our algorithm performs against other approaches on specific datasets, rather than simply verifying the theoretical guarantees.
>
> * As mentioned earlier, the only metric considered in our experiments is 0-1 classification loss. For the experiment setup, as described in Appendix F, we randomly split each dataset into training and testing sets, whose sizes are $2/3$ and $1/3$ of the original dataset. Since LogReg, Lin, Rbf, XGB, and Forest are deterministic algorithms, we only ran one trial of each. However, due to the resource limitations we highlighted in Appendix F, we needed to randomly sub-sample the training set when running the robust list learning algorithm (Algorithm 4). This  introduces variability when running the Sparse and PERS algorithms. Thus, we ran 50 trials for each of these two algorithms, and reported the median of the 50 observed losses in Table 2. The 0-1 classification losses of XGB and PERS both have a * mark on HYPO because the * mark on the loss of XGB indicates it significantly outperforms PERS ont his benchmark, while the * mark on the loss of PERS indicates it significantly improves over the ERM sparse classifier. Again, we added the * mark on the first five baselines (LogReg, Lin, Rbf, XGB, Forest) if and only if they significantly outperform PERS, while we added * mark on PERS if and only if PERS significantly outperforms Sparse.
>
> We again thank the reviewer for the suggestions and will incorporate the suggested changes in upcoming revisions.
> # Questions
>
> * The reason we used UCI datasets is because our work is based on [1], which followed the line of works [2, 3], and these works used the exact same datasets as mentioned in Section 4.2. Also, verifying distributional properties for any given data sample is, in general, very challenging. In fact, there is a current line of work on learning representations while verifying the distributional assumptions called testable learning[4, 5, 6], which is of course relevant and helpful, but beyond the scope of our present work. For the experiment setting, please see our response to the second weakness.
>
> __Please see the common answer for the references__

---

### Author Response · Authors · 2025-11-18
**Reference**

[1]Huang, J., \& Juba, B. (2025). Distribution-Specific Agnostic Conditional Classification With Halfspaces. The 13th International Conference on Learning Representations.

[2]Wiener, Y., \& El-Yaniv, R. (2011). Agnostic selective classification. Advances in neural information processing systems, 24.

[3]Wiener, Y., \& El-Yaniv, R. (2015). Agnostic pointwise-competitive selective classification. Journal of Artificial Intelligence Research, 52, 171-201.

[4] Diakonikolas, I., Kane, D., Kontonis, V., Liu, S., \& Zarifis, N. (2023). Efficient testable learning of halfspaces with adversarial label noise. Advances in Neural Information Processing Systems, 36, 39470-39490.

[5]Gollakota, A., Klivans, A. R., \& Kothari, P. K. (2023, June). A moment-matching approach to testable learning and a new characterization of rademacher complexity. In Proceedings of the 55th Annual ACM Symposium on Theory of Computing (pp. 1657-1670).

[6]Diakonikolas, I., Kane, D., Liu, S., \& Zarifis, N. (2024, June). Testable learning of general halfspaces with adversarial label noise. In The Thirty Seventh Annual Conference on Learning Theory (pp. 1308-1335). PMLR.

[7]Diakonikolas, I., Kontonis, V., Tzamos, C., \& Zarifis, N. (2022, June). Learning general halfspaces with adversarial label noise via online gradient descent. In International Conference on Machine Learning (pp. 5118-5141). PMLR.

---

### Meta-Review · Area_Chair_yAbM · 2026-01-13

**Summary:**

**Summary** This paper studies personalized prediction, a learning scheme where a dedicated classifier is learned for each query point rather than a single universal model. The authors propose learning sparse linear classifiers on subpopulations defined by homogeneous halfspaces, aiming to improve interpretability while maintaining accuracy. The main contribution is a PAC-learning algorithm that achieves an O(opt^{1/4}) error bound under well-behaved distributions, combining a projected gradient descent method for learning reference classes with a robust list learner for sparse linear classifiers. The authors evaluate their approach on several UCI medical datasets, comparing against standard baselines.

**Evaluation** This submission received 3 reviews with mixed feedback (6, 4, 2). Reviewers acknowledged that the paper is well-written, technically sound, and addresses a new problem formulation. On the other hand, reviewers raised concerns about potential fit, incremental novelty over prior work, experimental evaluation with unclear connections to theory, and limited practical insight from the proposed formulation. Having read the reviews and rebuttal, I am unfortunately recommending rejection. My decision is mostly motivated by the feedback from Hoge and 8waY. Even as the rebuttal did address some of the initial concerns regarding the work, these specific concerns remain unresolved, particularly the need broader applicability.  I encourage the authors to provide concrete examples or case studies demonstrating when personalized prediction succeeds where traditional PAC learners fail, clarifying the practical value of the approach.

**Reviewer Concerns:**

Outstanding:

- Questions about Significance [Hoge, 8waY, G5E4]
- Missing Diagnostic Validation [Hoge]

Resolved/Ignored:

- Incremental Novelty [G5E4]

**Reviewer Scores:**

I expect no major changes: Hoge would have remained at 4 since the rebuttal did not address experimental clarity, G5E4 would have remained at 6 or increased to 7 given the clarification on algorithmic novelty, and 8waY would have remained at 2.

---

### Decision · Program_Chairs · 2026-01-26

Reject